# Selection of HIV-1 for resistance to fifth-generation protease inhibitors reveals two independent pathways to high-level resistance

Ean Spielvogel[1,2], Sook-Kyung Lee[2], Shuntai Zhou[2], Gordon J Lockbaum[3†], Mina Henes[3‡], Amy Sondgeroth[1,2], Klajdi Kosovrasti[3], Ellen A Nalivaika[3], Akbar Ali[3], Nese Kurt Yilmaz[3], Celia A Schiffer[3*], Ronald Swanstrom[2,4*]

[1]Department of Microbiology and Immunology, University of North Carolina at Chapel Hill, Chapel Hill, United States; [2]Lineberger Comprehensive Cancer Center, University of North Carolina at Chapel Hill, Chapel Hill, United States; [3]Department of Biochemistry and Molecular Biotechnology, University of Massachusetts Medical School, Worcester, United States; [4]Department of Biochemistry and Biophysics, University of North Carolina at Chapel Hill, Chapel Hill, United States

**\*For correspondence:**
celia.schiffer@umassmed.edu (CAS);
ron_swanstrom@med.unc.edu (RS)

**Present address:** †Accent Therapeutics, Lexington, United States; ‡Emory University School of Medicine, Atlanta, United States

**Competing interest:** The authors declare that no competing interests exist.

**Abstract** Darunavir (DRV) is exceptional among potent HIV-1 protease inhibitors (PIs) in high drug concentrations that are achieved in vivo. Little is known about the de novo resistance pathway for DRV. We selected for resistance to high drug concentrations against 10 PIs and their structural precursor DRV. Mutations accumulated through two pathways (anchored by protease mutations I50V or I84V). Small changes in the inhibitor P1'-equivalent position led to preferential use of one pathway over the other. Changes in the inhibitor P2'-equivalent position determined differences in potency that were retained in the resistant viruses and that impacted the selected mutations. Viral variants from the two pathways showed differential selection of compensatory mutations in Gag cleavage sites. These results reveal the high level of selective pressure that is attainable with fifth-generation PIs and how features of the inhibitor affect both the resistance pathway and the residual potency in the face of resistance.

## Editor's evaluation

This work provides a fundamental understanding of the evolution of resistance to Darunavir, an exceptionally potent HIV-1 protease inhibitor. Conclusions are supported by compelling evidence that small changes in the inhibitor lead to two resistance pathways, each anchored by a specific mutation. These results provide the first evidence for de novo pathway selection and provide an atomic basis for designing the next generation of HIV-1 protease inhibitors.

## Introduction

Highly active antiretroviral therapy against HIV-1 with combinations of drugs effectively block viral replication and preclude the evolution of drug resistance. Three major factors interplay to define the emergence of resistance in vivo: (i) the active drug concentration relative to its inhibitory activity; (ii) the level of resistance conferred by one or more mutations; and (iii) the fitness cost of the resistance mutations. For protease inhibitors (PIs), the number of mutations selected increases with increasing

drug concentration (*Watkins et al., 2003*). Thus, the concentration of this drug class in vivo is an important parameter in determining the genetic barrier to resistance.

The early identification of the retroviral protease as a member of the aspartyl proteinase family and the determination of a number of cleavage site sequences led to the development of first-generation PIs (PIs) that validated the HIV-1 protease as a drug target (*Katoh et al., 1987*; *Richards et al., 1989*; *Seelmeier et al., 1988*). A second generation of PIs was quickly developed for use in humans, becoming the third drug in a three-drug regimen that achieved sustained suppression of viral load with no evolution of resistance (*Gulick et al., 1997*). The third generation of PIs had improved properties with regard to side effects and efficacy. These properties have been further enhanced with a fourth-generation PI, darunavir (DRV), which achieves a drug level in plasma (>1 μM) that is 1000-fold greater than its inhibitory activity in cell culture (*Ali et al., 2010*; *Kurt Yilmaz et al., 2009*; *Nalam et al., 2013*). The potential efficacy of a fourth-generation PI such as DRV has led to attempts to use this drug in monotherapy (*Antinori et al., 2015*; *Arribas et al., 2012*; *Paton et al., 2015*; *Valantin et al., 2012*). Although the initial trials of monotherapy showed a modest increase in the loss of viral suppression relative to standard ART, in the cases of virological failure there was no significant resistance to DRV detected in the rebound virus (*Antinori et al., 2015*; *Arribas et al., 2012*; *Paton et al., 2015*; *Valantin et al., 2012*). Thus, the observed virological failure is most easily attributed to issues of low drug exposure in specific participants. While therapy has not yet moved to single agents, potent two drug therapies are being explored. In this regard, several studies have shown that a PI can be part of a successful two drug therapy regimen (*Casado et al., 2020*; *Di Cristo et al., 2020*; *Hawkins et al., 2019*; *Huang et al., 2019*; *Vizcarra et al., 2020*).

Drug resistance to DRV is incompletely understood. The appearance of DRV-resistant viruses during therapy failure has occurred in people who had previously experienced virological failure with other PIs (*de Meyer et al., 2008*; *Descamps et al., 2009*; *Pellegrin et al., 2008*; *Tremblay, 2008*). In these cases, the pathway to DRV resistance was likely influenced by the preexisting resistance to the previous PI. DRV resistance has been selected in cell culture (*Aoki et al., 2018*; *de Meyer et al., 2008*; *Koh et al., 2010*). High-level resistance was selected starting with either a mixture of highly resistant variants that were allowed to recombine or the selection was started with a single PI-resistant strain; in both cases, an I84V resistance pathway in the protease was observed. However, there are no reports of selection of high-level resistance starting with an unmutated 'wildtype' virus. Thus, there is little information on what a do novo pathway for resistance to DRV would be.

An important concept in HIV-1 PI design is to avoid chemical moieties that extend outside of the substrate envelope, the shared volume in the protease active site that is occupied by natural protease substrates (*King et al., 2004*; *King et al., 2002*). We have previously designed a series of highly potent PIs based on DRV, UMASS-1 through -10, that still fit within the substrate envelope (*Nalam et al., 2013*). The designed fifth-generation inhibitors have modified chemical moieties that relative to DRV further fill the substrate envelope at the equivalent of the P1' and P2' positions (*Figure 1*). All bind tightly to the wild-type HIV-1 protease with a $K_i$ of less than 5 pM. These inhibitors retained robust binding to many multidrug-resistant protease variants and viral strains.

In an effort to define the de novo pathway to resistance for DRV and determine the effects of these chemical changes in the inhibitors, we examined the evolutionary path used to attain high-level resistance to DRV and the UMASS-1 through -10 panel of PIs. Under continuous and escalating selective pressure, representing between 50 and 95 passages, the virus evolved to accumulate multiple mutations. In most cases, selection was carried out until the final concentration approximated that achieved by DRV in vivo. Relatively minor modifications in inhibitor structure favored selection of one of two pathways to resistance, anchored by protease mutation I50V or I84V. In addition, the P2'-equivalent chemical structure of the inhibitor affected residual potency against highly resistant strains and in some cases changed the potency of the P1'-equivalent moiety. These results reveal the extremely high genetic barrier to resistance to the fourth-generation PI DRV at inhibitor concentrations that can be achieved in vivo, and the complex evolutionary pathways required to achieve resistance. In addition, these results reveal features of fifth-generation protease inhibitor design that both affect the selected pathway of resistance and determine residual potency against resistant variants.

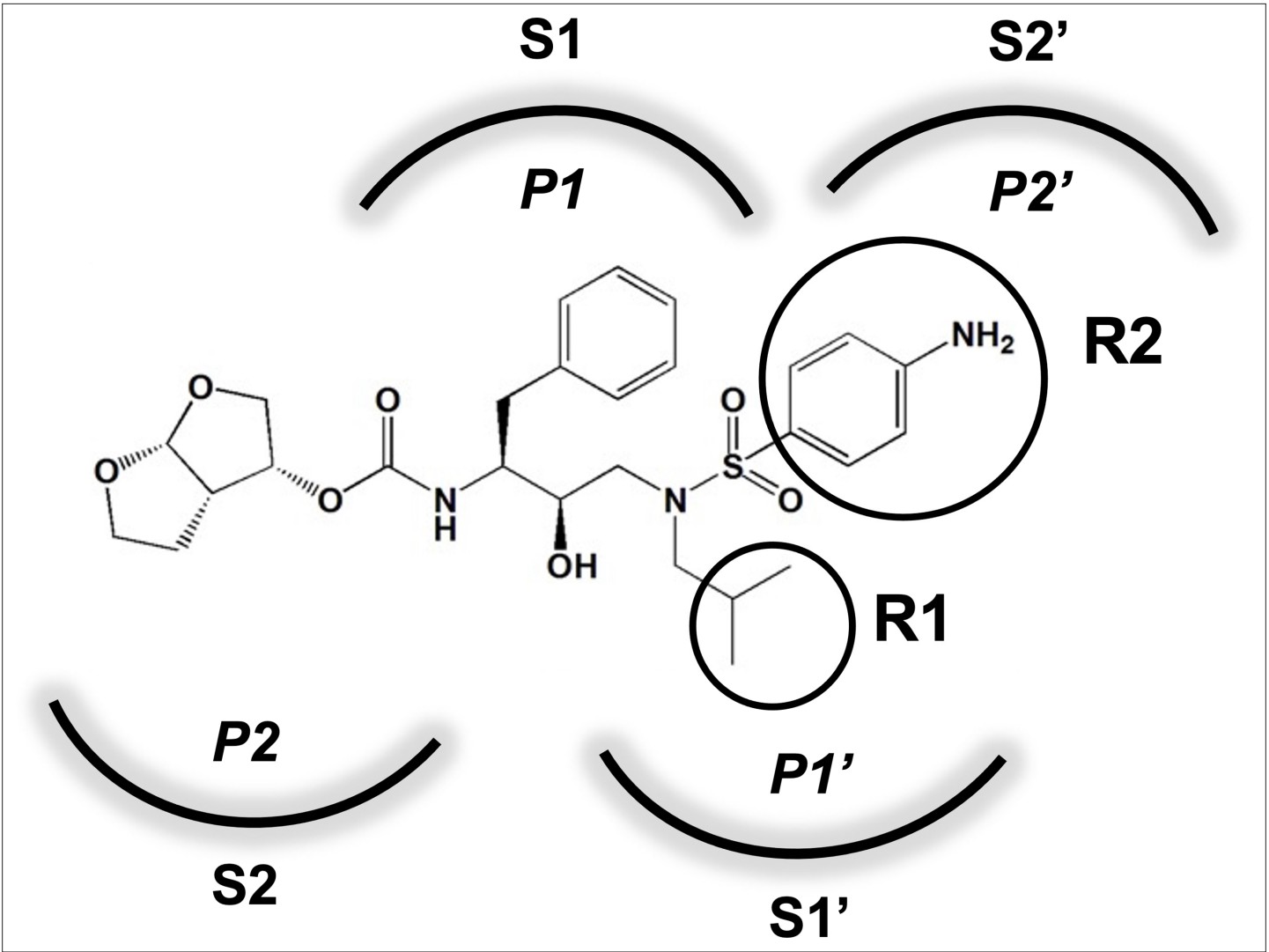

**Figure 1.** Darunavir (DRV) structure as a PR substrate analog. The chemical structure of DRV is shown. Protease substrates are labeled following a convention where the scissile bond is flanked upstream by amino acids labeled P1, P2, etc., with the amino acids downstream labeled P1', P2', etc. The cleavage site can thus be written P1/P1'. The side chains of these amino acids extend into subsites in the protease with corresponding labels (i.e., the P1 side chain extends into the S1 subsite). In the UMASS series of analogs, the DRV structure at the equivalent of the P1' position (labeled R1) was modified as was the structure at the equivalent of the P2' position (labeled R2).

## Results

### Panel of highly potent and analogous HIV-1 PIs

HIV-1 PIs were designed by modifications to DRV to increase favorable interactions with the protease within the substrate envelope, thereby increasing potency while minimizing evolution of resistance (*Nalam et al., 2013*). A panel of 10 DRV analogs were chosen with enzymatic inhibition constants ($K_i$) in the single or double-digit picomolar range to wild-type NL4-3 protease and the I84V and I50V/A71V drug-resistant variants, respectively (*Table 1*; *Lockbaum et al., 2019*; *Mittal et al., 2013*). These PIs contained modified P1' positions with (S)-2-methylbutyl or 2-ethyl-n-butyl moieties (R1-1 and R1-2, respectively) in combination with five diverse P2' phenyl-sulfonamides (R2-1 to R2-5), with the inhibitors named UMASS-1 through -10 (*Table 1*). These inhibitors and DRV were also tested in a cell culture-based viral inhibition assay. The $EC_{50}$ values (the effective concentration needed at the time of virus production to reduce infectivity by 50%) for DRV and the UMASS analogues ranged from 2.4 to 9.1 nM, significantly more potent than the second- and third-generation PIs tested (*Figure 2—figure supplement 1*).

**Table 1.** Structures, $K_i$ constants, and $EC_{50}$ values of the UMASS series of HIV-1 protease inhibitors (PIs).

| Inhibitor | Structure | $K_i$ (pM) | | | $EC_{50}$ (nM) | |
| | | WT | I84V | I50V/A71V | WT | Viral culture (5000 nM) |
|---|---|---|---|---|---|---|
| DRV | | <5.0* | 25.6 ± 5.6* | 74.5 ± 5.6* | 7.7 ± 1.6 | >100,000 |
| UMASS-1 | | <5.0* | 26.1 ± 3.7* | 110.3 ± 8.8* | 5.9 ± 1.0 | |
| UMASS-2 | | <5.0 | <5.0 | 15.0 ± 2.7 | 2.4 ± 0.3 | |
| UMASS-3 | | <5.0 | 9.9 ± 2.7 | 79.9 ± 5.9 | 9.1 ± 1.0 | 14,800 ± 6800 |
| UMASS-4 | | <5.0 | 10.5 ± 1.8 | 32.9 ± 3.0 | 3.2 ± 0.4 | 13,700 ± 6600 |
| UMASS-5 | | <5.0 | 7.0 ± 1.7 | 7.8 ± 0.9 | 4.0 ± 0.5 | >100,000 |
| UMASS-6 | | <5.0* | 12.8 ± 3.1* | 100.0 ± 9.9* | 5.2 ± 0.8 | |
| UMASS-7 | | <5.0 | 12.1 ± 4.5 | 18.2 ± 3.0 | 3.1 ± 0.5 | |
| UMASS-8 | | <5.0 | <5.0 | 55.4 ± 4.0 | 4.2 ± 0.9 | |
| UMASS-9 | | <5.0 | 7.6 ± 1.6 | 42.3 ± 2.6 | 6.4 ± 1.2 | >100,000 |
| UMASS-10 | | <5.0 | 14.3 ± 9.3 | 5.8 ± 1.1 | 4.1 ± 0.9 | |

*Previously reported in Lockbaum et al. (ACS Infect Dis. 2019 Feb 8; 5 (2): 316–325).

There are several patterns worth noting in the $K_i$ and $EC_{50}$ values. First, all inhibitors had a $K_i$ of less than 5 pM (limit of detection) against the wild-type enzyme, precluding a comparison of potency among them without using partially resistant mutant proteases (I84V and I50V/A71V). Second, the P1'/R1 position change to the larger 2-ethyl-n-butyl moiety (R1-2, UMASS-6) increased potency against the I84V mutant protease relative to DRV and the intermediate sized R1-1 moiety UMASS-1 (all three share the same R2-1 moiety); this enhanced potency of the R1-2 inhibitors over the R1-1 inhibitors was also apparent for UMASS-8 over UMASS-3, which share the same R2-3 structure (*Table 1*, *Figure 2—figure supplement 2*), while this pattern of potency was reversed (i.e., R1-1 was more potent than R1-2) for UMASS pairs -2 and -7, and -5 and -10 (with R2-2 and R2-5, respectively; *Figure 2—figure supplement 2*). Third, four of the inhibitors with the R2 moieties R2-2 or R2-5 were significantly more potent (UMASS-2, -5, -7, -10) than with the other R2 moieties. This trend was most pronounced for activity against the I50V/A71V mutant enzyme in both the R1-1 and R1-2 backgrounds (*Figure 2—figure supplement 2*). The significant potency of the R2-2 and R2-5 containing inhibitors against the I50V/A71V mutant enzyme while maintaining good potency against the I84V enzyme indicates that both the R1 and R2 constituents contribute to potency, and to enhancing potency over DRV. This potency was also seen in inhibition of viral infectivity as the $EC_{50}$ values of the R2-2 and R2-5 containing inhibitors were among the lowest measured (*Table 1*).

## Selection for high-level resistance during passage in cell culture follows two pathways

To evaluate the potential of each inhibitor to select for mutations that would confer high-level resistance and to compare these mutations across the inhibitor series, we grew HIV-1 under conditions of escalating inhibitor concentration in cell culture through a lengthy period of selection (50–95 passages). The selection experiments were performed under two separate starting conditions: first when the starting virus was generated from the NL4-3 clone (this clone closely approximates the clade B consensus for the protease amino acid sequence and will be referred to as 'wild type'), then again when the starting virus was a mixture of 26 isogenic viruses each with a single mutation associated with drug resistance in the NL4-3 background. Notably, in the latter case only about one-half of the mutations that appeared in the culture during selections were in the mixture of starting mutations, indicating that even in the selections that were seeded with the pool of single resistance mutations there was sufficient evolutionary capacity to explore mutations at positions that started as the wild-type sequence. The initial inhibitor concentration started at low nanomolar concentrations and increased by a factor of 1.5 with each subsequent viral passage (the inhibitor concentration was increased only after the virus spread efficiently through the culture). All of the selections starting with wild-type virus reached at least 5 µM of inhibitor concentration (approximating the therapeutic concentration reached by DRV in the blood and in the range of a 1000-fold increase in the starting $EC_{50}$). For technical reasons, only five of the selections starting with the mixture of mutants reached an inhibitor concentration as high as 400 nM and are reported here (*Figure 2—figure supplement 3*).

Resistance mutations selected in the protease coding domain during the escalating selective pressure of increasing PI concentration were examined at various timepoints using a next-generation sequencing (NGS) protocol that included Primer ID with the MiSeq platform (*Zhou et al., 2015*). We first examined the sequence of the most abundant variant in each of the two selection schemes present at the highest inhibitor concentration reached (*Figure 2A*). For those selections that reached 5 µM, 8–14 mutations were present in the most abundant variant. Typically the next two most abundant variants in the population differed by a single amino acid, and on average the three most abundant variants in the 5 µM cultures accounted for 88% of the total viral population (*Figure 2—source data 1*). Evolution of resistance followed two pathways, one based on I50V and one based on I84V. In three cultures, V2wt, V5wt, and V2mut (which reached either 4 or 5 µM final drug concentration; V2wt indicates the virus pool selected with UMASS-2 starting with wild-type virus), these two mutations, I84V and I50V, became linked on the same genome (*Figure 2A*) showing they are not mutually exclusive; it is worth noting that the appearance of the linked I84V/I50V mutations occurred in the selections of the UMASS inhibitors (-2 and -5) that showed the greatest potency against the two mutant enzymes I84V and I50V/A71V (*Table 1*, *Figure 2—figure supplement 2*), consistent with inhibitor potency driving the co-selection of these two primary resistance mutations together.

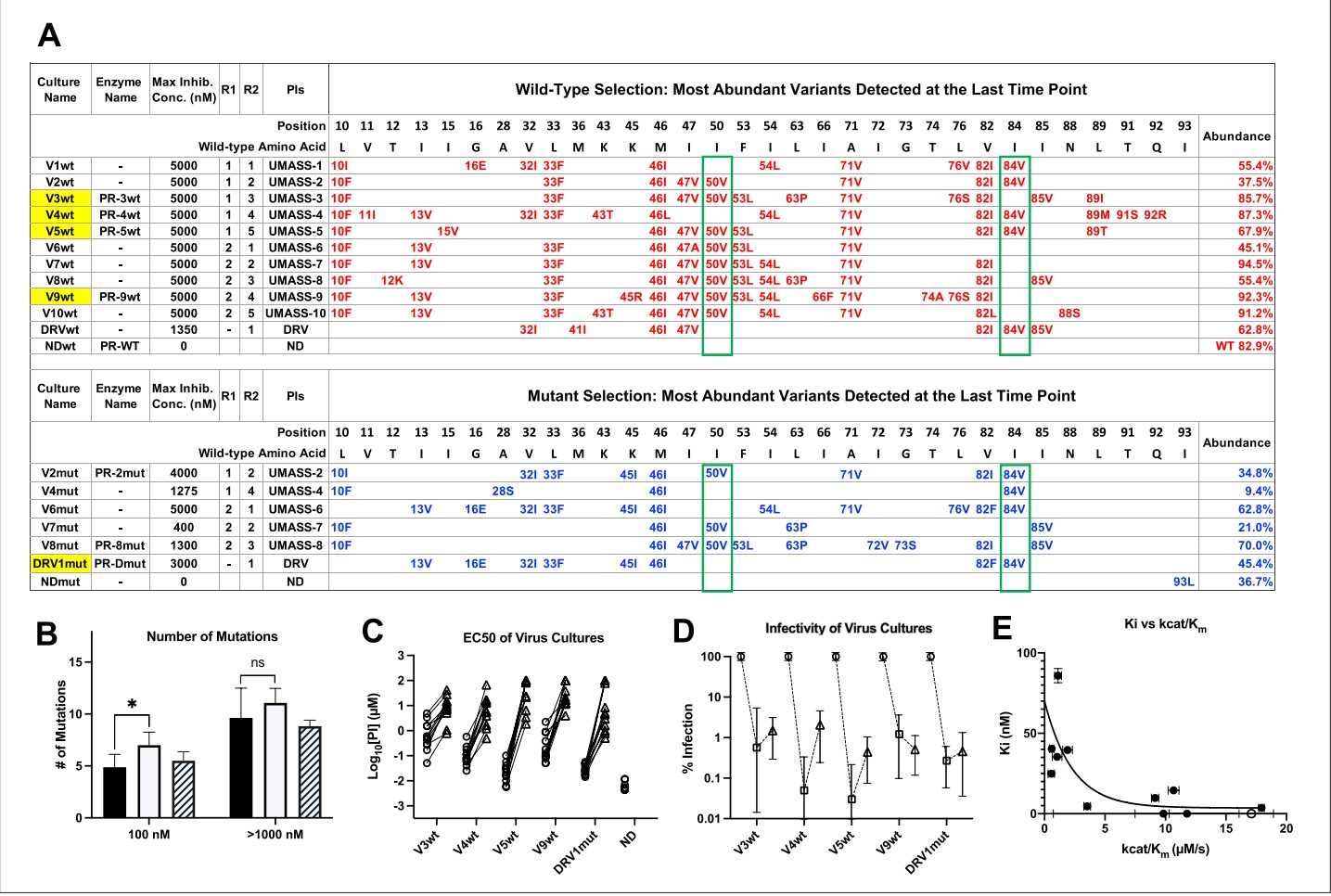

**Figure 2.** Features of viruses selected to high-level resistance against darunavir (DRV) and the UMASS-1 through -10 inhibitor series. Infected cultures were put under drug selection starting with either wild-type virus or a pool of mutant viruses. (**A**) Most abundant variants present at final timepoint. Resistance mutations in the viruses selected starting with the wild-type virus are in blue, and the selection starting with the pool of mutant viruses in in red. The culture name indicates a virus pool (V), the UMASS PI number (V1), and whether the selection started with wild-type or mutants (V1wt). PR sequences that were used to generate purified recombinant protease are indicated (e.g., PR-3wt). The final protease inhibitor (PI) concentration reached and the R1 and R2 moieties are indicated along with the PI. (**B**) Number of mutations present in the three most abundant variants of each selection at two different PI concentrations. Cultures containing I84V resistance mutation are in black (n = 4), I50V-containing cultures are in white (n = 7), and cultures with I50V+I84V linked are shown with hatched box (n = 3). Unpaired $t$-test was used to compare the number of mutations in the 84V vs. the 50V cultures. (**C**) $EC_{50}$ values for highlighted selections in panel (**A**) reaching 100 nM (square) and 5000 nM (triangle). (**D**) Relative infectivity values for the same selections shown relative to the wild-type virus (circle). Relative infectivity was measured with normalized input amounts of HIV-1 p24 CA protein in replicates (n=3). Error bars show range of values. (**E**) Enzyme inhibition constants ($K_i$) of end point PR variants versus catalytic efficiency (kcat/$K_m$). Open circle represents wild-type NL4-3. Trendline is for visualization purposes only. * p<0.05, **p<0.01, ***p<0.001.

The online version of this article includes the following source data and figure supplement(s) for figure 2:

**Source data 1.** Source sequence data for top abundant variant in panel A.

**Source data 2.** Source sequence data for top 3 most abundant variants in panel A.

**Figure supplement 1.** $EC_{50}$ inhibition curves for 2nd, 3rd, 4th, and 5th generation protease inhibitors.

**Figure supplement 2.** Enzymatic inhibition constant ($K_i$) values for UMASS-1–10 in the presence of pro with I84V or I50V/A71V mutations.

**Figure supplement 3.** Increasing inhibitor concentrations of viral selection passages.

**Figure supplement 4.** Shannon's entropy in the protease region through five of five darunavir (DRV) selections.

**Figure supplement 4—source data 1.** Source sequence data for top abundant variant DRV passages in panel A.

In addition to the sequence of the terminally selected virus, we validated that selection occurred over the entire course of the selection protocol. First, as can be seen in *Figure 2B*, the number of mutations present at 100 nM drug concentration was approximately half the number present after selection to greater than 1 µM. In comparing the number of mutations present in cultures using the I84V versus the I50V pathway, we noted that on average there were more mutations in the I50V viruses at the intermediate drug concentration compared to the I84V viruses (100 nM, p=0.03), and this trend continued at the higher drug concentration. Next we compared the $EC_{50}$ of five virus cultures representing the I84V or the I50V pathways (highlighted in yellow in *Figure 2*), and including one that had both mutations linked, testing the level of resistance to all of the UMASS-1 through -10 inhibitors after selection to 100 nM drug concentration or to 5 µM. As can be seen in *Figure 2C*, resistance (measured as an increase in the $EC_{50}$) was apparent after selection to the 100 nM inhibitor concentration, and in each case the $EC_{50}$ increased an average of 100-fold after selection to the final inhibitor concentration of 5 µM ($EC_{50}$ values of the final virus pool are shown in *Table 1*).

We were also interested in the fitness cost of these mutations. To measure this effect, we compared the infectivity of the virus in 10 culture supernatants (five virus pools at each of two levels of selection), normalizing infectivity on a reporter cell line for a given amount of the virion p24 CA protein in the culture supernatant. This was compared to the unmutated NL4-3 parent (given a value of 100%). As can be seen in *Figure 2D*, selection to 100 nM inhibitor concentration resulted in a reduction in fitness/relative infectivity by between 100- and 1000-fold. Those cultures with the biggest decrease in fitness rebounded to around 1% relative infectivity after selection to an inhibitor level of 5 µM. This suggests that these high levels of resistance are linked to maintaining a residual level of relative infectivity around 1% (as defined under these culture conditions).

We used another approach to explore the interplay between resistance and fitness; we examined the effect of resistance on catalytic efficiency of the protease by measuring $K_i$ and the catalytic efficiency ($kcat/K_m$) for a set of mutant proteases against the entire panel of UMASS inhibitors and DRV. The average $K_i$ value of each mutant protease (sequences shown in *Figure 2—source data 1*) is plotted against the catalytic efficiency ($kcat/K_m$) for that enzyme in *Figure 2E*. As can be seen, there is a strong relationship between higher $K_i$ values (i.e., resistance) and dramatic reductions in catalytic efficiency, consistent with the loss of fitness associated with these viral populations.

## Features of the selection process assessed by NGS of longitudinal samples

Each of the viral cultures started with wild-type virus showed an accumulation of protease mutations with increasing selective pressure. NGS analysis revealed very few fixed variants in the cultures started with wild-type virus until the inhibitor concentration reached approximately 3–5 nM (approaching the $EC_{50}$ value for the wild-type virus, *Table 1*); in contrast, the cultures that were started with the mutant library selected for the outgrowth of a subset of those mutants by 1 nM inhibitor concentration (*Figure 3A*). In all cultures, the transition through the $EC_{50}$ concentration of the wild-type virus provided significant selective pressure for fixing mutations. Multiple unfixed mutations were observed in each culture after the drug concentration exceeded the $EC_{50}$ values above 3 nM, highlighting the high genetic diversity in the culture. Additional mutations became linked on each viral genome at higher drug concentrations.

Deep sequencing revealed that mutations accumulated in complex patterns. We assessed the sequence complexity of each culture by calculating the Shannon entropy to allow comparison of changes in the diversity in each culture over time and as a function of increasing selective pressure (*Figure 3A*). In the cultures that showed the early appearance of the I84V mutation, this was associated with a peak in entropy, reflecting high genetic diversity, followed by a decrease in entropy when the I84V mutation became fixed (*Figure 3—figure supplement 1A*). The introduction of the I50V mutation was generally not associated with a drop in entropy, rather these populations maintained high genetic diversity even at higher drug concentrations (*Figure 3—figure supplement 1B*). We interpret these patterns as indicative of I84V conferring some level of resistance without a dramatic loss in fitness, allowing a more homogeneous culture (i.e., less entropy). In contrast, I50V may confer a higher level of resistance but at a greater fitness cost, thus supporting greater diversity in the culture either as compensatory mutations or as other combinations of mutations with lesser resistance but higher fitness. In this regard, we previously showed I50V significantly reduces the

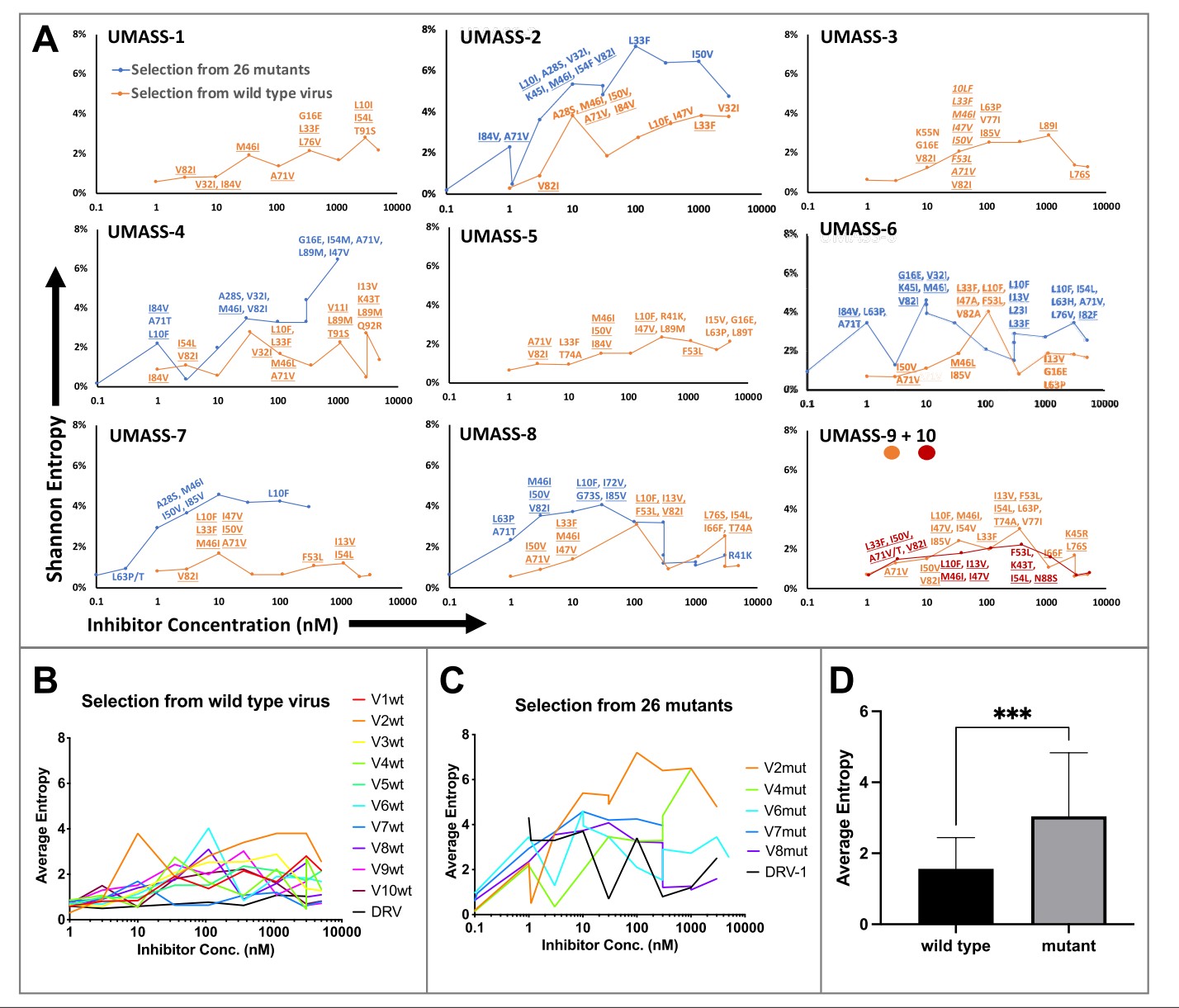

**Figure 3.** Evaluation of sequence diversity in the viral population during increasing selective pressure with protease inhibitor (PI) inhibitors. (**A**) Shannon entropy was calculated using sequence diversity in the protease region. Abundant mutations are shown and fixed mutations are underlined. Selections from wild type are in orange (n = 6) and from the mutant pool (n = 11) in blue. Two selections from wild-type virus are shown in orange (UMASS-9) and red (UMASS-10). (**B**) Compiled Shannon entropy values from selections starting with wild-type virus that reached concentrations of >1000 nM. (**C**) Compiled Shannon entropy values from selections starting with the pool of mutant viruses that reached concentrations of at least 400 nM. (**D**) Selections starting with the pool of 26 mutants showed higher entropy than the pool starting with WT (n=10, n=10) when averaged over all analyzed drug concentrations (p<0.001 using the unpaired *t*-test). Error bars show standard deviation.

The online version of this article includes the following figure supplement(s) for figure 3:

**Figure supplement 1.** Shannon's entropy of viral cultures undergoing difference resistance pathways.

**Figure supplement 2.** Abundance of I50V and I84V mutations at each drug concentration of all UMASS inhibitors that reached at least 1000 nM concentrations in culture.

fitness of the virus relative to the fitness loss of a virus with I84V as single mutations (*Henderson et al., 2012*).

When we examined the sequence diversity as assessed by Shannon entropy for all selections that reached at least 1 µM in inhibitor concentration, we found that cultures starting with the mixture of

resistant viruses averaged a nearly twofold higher entropy value compared to the cultures where the selection started with just the virus generated from the NL4-3/wild-type clone (3.0 vs. 1.6, p<0.0001 Mann–Whitney test; *Figure 3B–D*). This was unexpected as both sets of selections passed through many genetic bottlenecks. This result is most easily explained if the rates of recombination were fairly high throughout the culture period so that most sequence variants were maintained at least at a low level. However, at the end of the selection period no wild-type variants could be detected in the cultures by deep sequencing.

## The chemical nature of R1 and R2 determines the resistance pathway

We next examined whether the inhibitor structure influenced the resistance pathway chosen. We found that the R1 group, that is, the (S)-2-methylbutyl (R1-1) or the 2-ethyl-n-butyl (R1-2), largely defined the resistance pathway observed; for these inhibitors, the R1 group takes the position of P1' in the protease substrate analog, occupying the S1' subsite. With the UMASS-1 through -5 series (the smaller R1-1 group), the I84V mutation appeared first in six of seven cultures. In contrast, the UMASS-6 through -10 series with the larger R1-2 group, the I50V mutation appeared first in eight of nine cultures (p=0.009, Fisher's exact test; in this analysis, we included two cultures that did not reach at least 400 nM inhibitor concentration but did fix an initial set of mutations to increase our sample size [V3mut, V5mut], and we did not include two cultures where both I84V and I50V were initially fixed together [V2wt, V5wt]). We considered the possibility that the starting mixture of viruses in the mutant selection might skew the pathway selected, especially since the mutant pool included I84V but not I50V. However, in only one of the eight cultures with sufficient data from both selections was there a switch from the I84V pathway to the I50V pathway between the first and second selections (cultures of UMASS-6 with an R1-2 group). Thus, we conclude that the P1'-equivalent chemical structure of the inhibitor is a strong determinant of the resistance pathway selected. It is notable that one inhibitor could select for different pathways in two separate selections (also seen with DRV, see *Figure 2—figure supplement 4*) even when both major mutations are maintained in the viral population (*Figure 3—figure supplement 2*). This suggests that either pathway can provide some level of resistance to most if not all of these inhibitors, and that the chance addition of the initial compensatory mutations may determine which pathway becomes the major resistant population.

To examine the potential for linked mutations and infer the order in which mutations accumulated in the protease gene to confer high-level resistance, the abundance data from multiple selections that ended in one or the other pathway were pooled and compared. In this analysis, shown in *Figure 4A*, summary data for the selections resulting in the I84V pathway point up, with I84V reaching 100% penetrance by definition. Similarly, the summary data for those selections that fixed I50V are shown pointing down, with I50V reaching 100% penetrance. Several mutations are uniquely linked or at least strongly favored in each pathway, with I84V being linked to V32I, and I50V being linked to I47V, F53L, and I13V. A number of mutations appear in both pathways, although not with equal frequency: L10F, L33F, M46I, I54L, A71V, and V82I. Finally, other mutations appear less frequently, making it difficult to assign them to one of these categories. These results show that while some mutations are largely linked to one pathway, other mutations are often shared between the two pathways. Also, the variation in frequency of appearance of shared mutations in the two pathways suggests different levels of impact on resistance and/or fitness in the I84V vs. the I50V background for these mutations (e.g., L10F, M46I and A71V). In particular, the larger impact of I50V on fitness compared to I84V mutations (*Henderson et al., 2012*) is consistent with the earlier appearance and greater levels of fixation of the shared compensatory mutations L10F, L33F, M46I, and A71V in the cultures that followed the I50V pathway. In *Figure 4B*, we show a summary timeline of the ordered addition of mutations for each pathway and whether they are shared or unique to the pathway (in addition to their relative penetrance/final prevalence among the different cultures). In *Figure 4—figure supplement 1*, we show phylogenetic trees of the viral protease sequence from these longitudinal selections annotated to show where in the tree each of the fixed final mutations entered the viral population. These results emphasize that there are stochastic elements in the selection process that make each culture different in detail but that by pooling the data across the cultures underlying patterns are apparent.

We next considered the possibility that the R2 constituent would provide additional selective pressure in the form of additional resistance mutations. We did not detect any novel mutations associated with any of the R2-1 through R2-5 inhibitors (*Figure 2*). However, as noted earlier, the inhibitors with

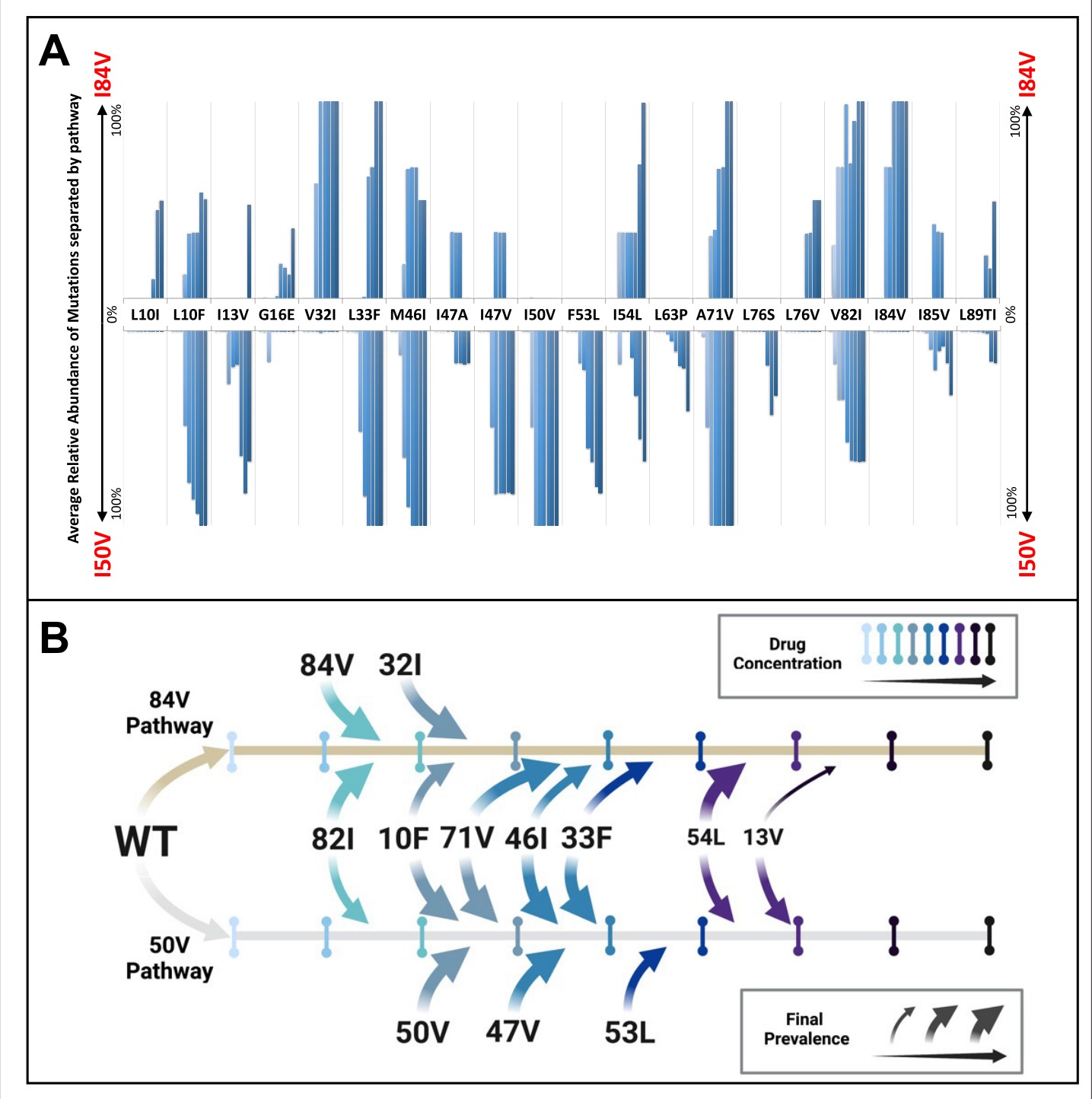

**Figure 4.** Patterns of accumulation of amino acid substitutions associated with resistance during selection. (**A**) Abundance data from selections for each indicated amino acid substitution were pooled and examined sequentially at different levels of drug concentration. Selections that reached the maximum concentration of at least 3000 nM were assessed longitudinally to allow for all timepoints to be averaged equally. Mutations from selections resulting in the I84V pathway (V1wt, V4wt, and V6mut) point up, with I84V reaching 100% penetrance by definition. Similarly, those selections that fixed I50V only (V3wt, V6-10wt) are shown pointing downward, with I50V reaching 100% penetrance. (**B**) Schematic representations of the addition of mutations found in I84V and I50V pathways. Each mutation's position on the timeline of drug concentrations was estimated using the pools of selections from each I84V and I50V pathways. This summary does not reflect the exact order and linkage of mutations for each individual selections in that pathway but rather the generalizable pattern inferred by comparing across the multiple cultures. The size of the arrows represents how prevalent that mutation was among the cultures at the end of the selection process.

*Figure 4 continued on next page*

*Figure 4 continued*

The online version of this article includes the following figure supplement(s) for figure 4:

**Figure supplement 1.** Introduction of mutations at increasing drug concentrations.

R2-2 and R2-5 in the smaller R1-1 background (UMASS-2 and UMASS-5) resulted in the selection of resistant virus where both of the primary resistance mutations (I84V and I50V) were present and linked. The comparable R2 inhibitors in the R1-2 background (UMASS-7 and UMASS-10) selected for just the I50V primary mutation consistent with the larger R1-2 structure. Thus, for inhibitors with three of the R2 moieties (R2-1, R2-3, R2-4) the selection of the primary resistance pathway was determined by the size of the R1 moiety. This was also true for the R2 moieties R2-2 and R2-5 with the larger R1-2 moiety, selecting for I50V; however, these same two R2 moieties are able to change the interaction of the smaller R1-1 moiety to drive selection of both primary resistance mutations demonstrating a role for R2 in influencing the resistance pathway.

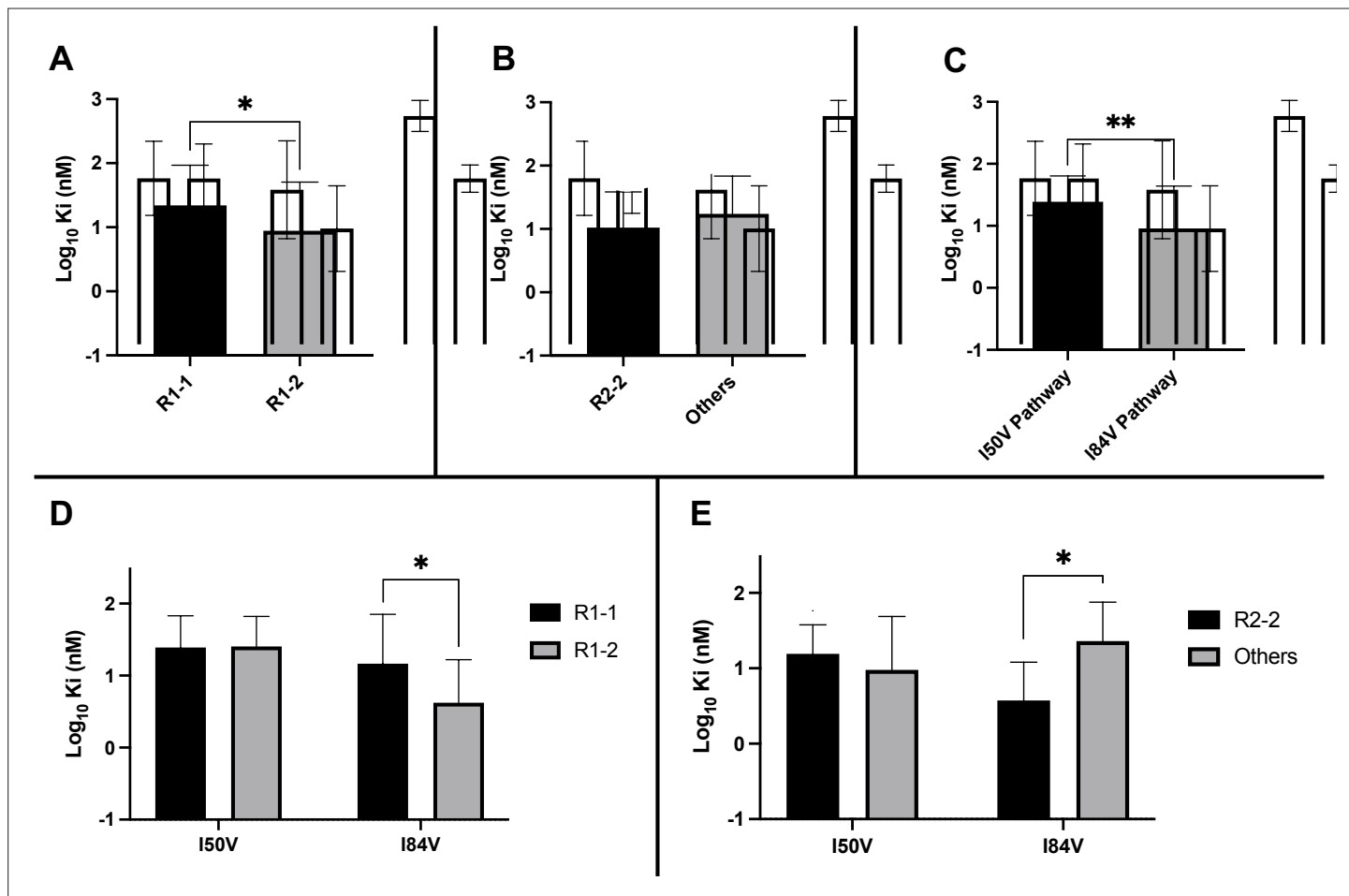

**Figure 5.** Analysis of $K_i$ values for mutant enzymes. $K_i$ values were determined against all UMASS inhibitors (**Table 1**). Brackets above the bars represent significant p-values between the two groups using the unpaired *t*-test. Data were pooled in different ways for the following analyses: (**A**) the $K_i$ values against the inhibitors with the larger R1-2 moiety (n = 34) were more potent against resistant proteases compared to the inhibitors with the smaller R1-1 moiety (n = 35) (**B**) $K_i$ values for the inhibitors with the R2-2 group (n = 10) showed a trend toward being more potent against the highly mutated proteases compared to the other R2 groups (n = 40). (**C**) $K_i$ values for enzymes with the I50V mutation (n = 33) showed greater resistance to the inhibitors compared to enzymes with the I84V mutation (n = 22). (**D**) The R1-2 moiety provided increased potency to enzymes with the I84V mutation (n = 20) but not the I50V mutation (n = 20). (**E**) The R2-2 moiety was more potent against the enzymes with I84V mutation (n = 10) compared to the enzymes with the I50V mutation (n = 10). The unpaired *t*-test was used to assess differences in $K_i$ values. Error bars show standard deviation.

### The chemical nature of R1 and R2 determines residual potency among the resistant variants

We considered several variables in examining the nature of the interaction between inhibitors and the resistant proteases: the extent to which the inhibitor structure (either R1 or R2) affected residual potency; and the extent to which the pathway (I84V or I50V) conferred the greatest resistance. As shown in *Figure 5A*, the inhibitors with the smaller R1-1 group showed a trend to be less potent than the inhibitors with the larger R1-2 group when tested against all of the resistant proteases. Inhibitors with the R2-2 group showed a trend toward greater residual potency against the mutant enzymes compared to the other inhibitors (*Figure 5B*). In looking at the individual pathways, the highly resistant enzymes with the I84V mutation remained more sensitive to the entire group of inhibitors than the enzymes with the I50V mutation (*Figure 5C*).

We next linked the two pathways to the specific structural features of the inhibitors. The resistant proteases with I50V had similar $K_i$ values to both the R1-1 and R1-2 inhibitors, while the proteases with I84V were more sensitive (lower $K_i$) to the larger R1-2 inhibitors (*Figure 5D*). Similarly, the increased potency of the R2-2 inhibitors over the rest of the inhibitors was seen against the enzymes carrying I84V but not those with I50V (*Figure 5E*). These results are consistent with the smaller R1-1 inhibitors selecting for the I84V pathway and the larger R1-2 inhibitors selecting for the I50V pathway, and with the R2-2 inhibitors adding additional potency that is retained even after selection for resistance, most apparent with the I84V pathway.

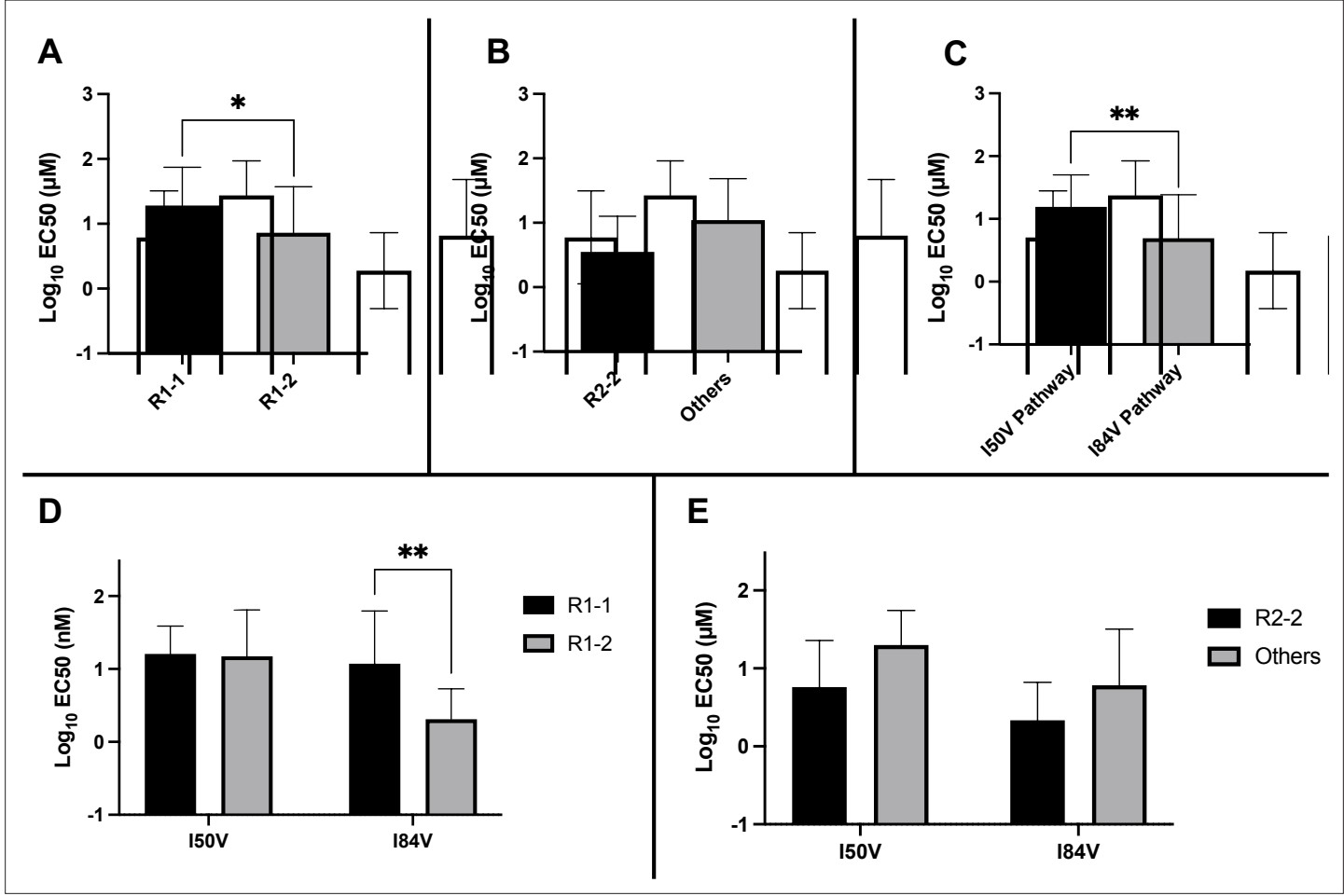

**Figure 6.** Analysis of $EC_{50}$ values for mutant virus cultures. $EC_{50}$ values were determined for a subset of the selected virus cultures against a panel of inhibitors (n = 50). For (**A–E**), the $EC_{50}$ data were pooled using the same methods and sample numbers as in *Figure 5*. The highest level of resistance recorded was 100 μM. The Mann–Whitney rank-sum test was used to assess differences in $EC_{50}$ values. Error bars show standard deviation.

Residual potency dependence of inhibitor structure could also be seen in the $EC_{50}$ values of the resistant virus pools. When the $EC_{50}$ values against the virus pools using the I84V or the I50V pathway were compared for all of the inhibitors, the inhibitors with the larger R1-2 group were more potent, retaining on average lower $EC_{50}$ values compared to the $EC_{50}$ values for the inhibitors with the smaller R1-1 group (*Figure 6A*). Inhibitors with the R2-2 group showed a trend toward greater potency compared to the other R2 groups (*Figure 6B*), consistent with what was observed in enzymatic assays. When we considered the effects based on the selection pathway, we observed that viruses in the I50V pathway had a higher level of resistance to these inhibitors than the viruses in the I84V pathway (*Figure 6C*). The higher level of resistance for the viruses using the I50V pathway was due to the fact that these viruses were similarly resistant to the inhibitors with either the R1-1 or the R1-2 groups; in contrast, the viruses using the I84V pathway conferred a greater level of resistance to the inhibitors with the smaller R1-1 group while the inhibitors with the larger R1-2 group retained a higher level of potency (*Figure 6D*). Finally, there was a trend for viruses in either pathway to be less resistant to the inhibitors with the R2-2 group (*Figure 6E*).

## DRV favors the I84V pathway

DRV has a butyl group at the R1 position, smaller than the R1-1 methylbutyl in the UMASS-1 through -5 series. We carried out five selections with DRV where the final inhibitor concentration reached greater than 1 µM. One DRV selection was carried out in parallel with each of the two different selections with the UMASS inhibitors (*Figure 2A*); both of these selections resulted in the appearance of the I84V mutation as defining the resistance pathway. Three additional selections were done in parallel with DRV and starting with the mixture of the 26 isogenic mutants; all three selections reached the level of 5 µM as the highest drug concentration. Two of these selections used the I84V pathway to resistance, while one of the selections used the I50V pathway to resistance (*Figure 2—figure supplement 4*). Thus, four of five independent selection with DRV favored the resistance pathway I84V, associated

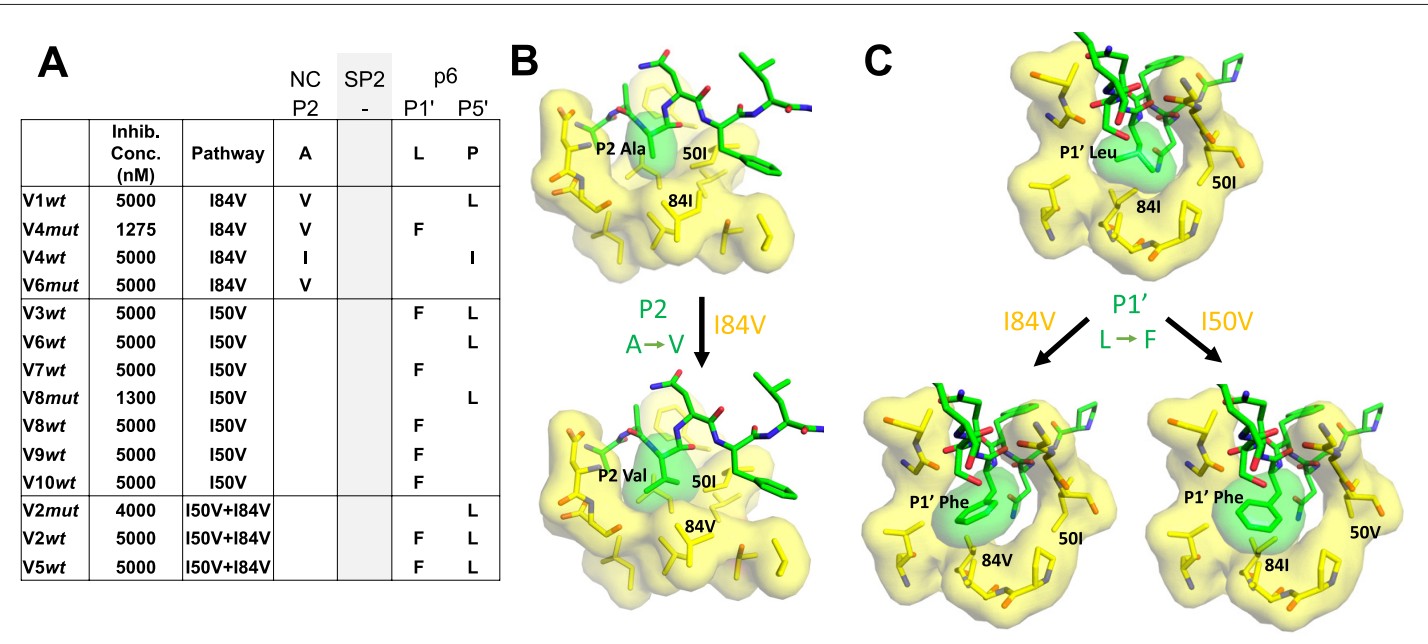

**Figure 7.** Protease cleavage site mutations observed after selection for high-level resistance. (**A**) Culture names, final inhibitor concentration, and resistance pathway are shown. Changes in amino acid sequence are shown for the NC/SP2 and SP2/p6 cleavage sites. (**B**) P2 substitution in the NC/SP2 cleavage site with the I84V mutation modeled. Subsite structure was modeled using the original structure (PDB: 1KJH) and mutating the P2 position in the subsite pocket from alanine to valine. (**C**) P1' change in the SP2/p6 cleavage site with either the I84V mutation or the I50V mutations modeled. Subsite structure was modeled using the original structure (PDB: 1KJF) and mutating the P1' position from leucine to phenylalanine.

The online version of this article includes the following source data for figure 7:

**Source data 1.** Sequenceing data for protease cleavage sites.

with a smaller R1 constituent. These results add further evidence for the size of the R1 moiety strongly influencing the resistance pathway.

## The I84V pathway and the I50V pathway differentially select for Gag cleavage site mutations

Cleavage site mutations are seen during selection for resistance to protease inhibitors (*Su et al., 2019*). This can be viewed as protease-substrate coevolution, and the effect is most apparent at the cleavage sites flanking the spacer peptide SP2 in Gag (NC/SP2 and SP2/p6) (*Kolli et al., 2006*; *Kolli et al., 2014*; *Kolli et al., 2009*; *Lee et al., 2012*; *Özen et al., 2011*; *Özen et al., 2012*; *Özen et al., 2014*; *Prabu-Jeyabalan et al., 2004*). We sequenced the protease cleavage sites encoded in the viral *gag* gene in the pools of selected viruses where the inhibitor concentration had reached a level of greater than 1 μM (*Figure 7A*). An analysis of four cultures that had I84V as the major resistance mutation showed they all had a mutation at the NC/SP2 cleavage site at position P2, with a change from the wild-type alanine amino acid to either of the larger aliphatic amino acids valine or isoleucine. In addition, three of the four I84V cultures had a mutation at the adjacent SP2/p6 cleavage site, either at P1' (leucine to phenylalanine) or P5' (proline to leucine). Conversely, all seven cultures where the I50V mutation was the major resistance mutation there was a mutation in the SP2/p6 cleavage site, but not in the NC/SP2 site. Finally, in the three cultures where the protease evolved both the I50V and I84V mutations, Gag mutations were observed only at the SP2/p6 cleavage site (the presence or absence of the NC/SP2 mutation in the I84V cultures but not in the I84V/I50V cultures has a p value of 0.03 in a Fisher's exact test). Two of three cultures with both I84V and I50V had both the P1' (leucine to phenylalanine) and P5' (proline to leucine) mutations. In contrast, mutations at both P1' and P5' together were underrepresented in the cultures with only I50V. An examination of the modeling (*Figure 7B*) suggests that the NC/SP2 mutation at P2 in the presence of the I84V mutation may engage I50 to replace the lost interaction with I84V as it moves away from the P2 sidechain by shortening (I to V); this would explain the absence of this cleavage site mutation in the double protease mutant (I50V/I84V) since the shortening of both protease sidechains provides too little interaction with the longer P2 sidechain. The P1' and P5' mutations in the SP2/P6 site appear to act in a complementary way as typically only one is seen with either the I50V or the I84V mutant proteases (*Figure 7C*). The SP2/p6 P1' and P5' mutations engage the protease by different mechanisms (*Özen et al., 2014*) suggesting their effects are additive; thus in these selections the single mutation at P1' or P5' may be sufficient to recover an appropriate rate of cleavage with either the I84V or the I50V mutant protease, but the additive effects of the P1' and P5' mutations may be needed to rescue cleavage by the I50V/I84V mutant protease.

## Discussion

DRV is a notable PI both for its tight binding to the HIV-1 protease and for its ability, with boosting doses of ritonavir, to reach micromolar levels of drug concentration in the blood. These features are important as resistance to PIs typically requires multiple mutations affecting both resistance and the rescue of enzyme activity/fitness lost with primary resistance mutations (*Arribas et al., 2012*; *Clemente et al., 2004*; *Henes et al., 2019*; *Mahalingam et al., 2002*; *Muzammil et al., 2003*; *Ragland et al., 2014*; *Wensing et al., 2010*). While we were successful in selecting for resistance to DRV in cell culture, the spectrum of mutations observed largely overlap those seen in previous studies based on testing individual mutations for their effect on DRV sensitivity (*de Meyer et al., 2008*; *King et al., 2004*; *Rhee et al., 2003*). In this work, we have explored chemical modifications to the DRV scaffold. We found that modification of the P1'/R1 chemical structure to be a larger aliphatic group favored the use of the I50V resistance pathway compared to the smaller DRV structure or an intermediate-sized structure, forcing the virus to use the more deleterious I50V mutation compared to the I84V mutation used with the smaller P1'/R1 structures. Conversely, we identified a subset of the P2'/R2 groups that contributed to residual potency even in the resistant proteases but did not select for novel resistance mutations; however, several of the R2 groups that conferred higher potency selected both major primary mutations but only with the smaller R1-1 methylbutyl group. Thus, chemical modifications at both of these inhibitor positions yielded improvements over DRV

and an unexpected linkage between the R1 and R2 groups in that anchoring the R2 group to the protease backbone enhanced the potency of the intermediate-sized R1 group, at least as assessed by its selective pressure.

The potency of these inhibitors can be inferred by the high genetic barrier to high-level resistance. The maximum drug concentrations of DRV achieved in the blood are nearly 1000-fold above the $EC_{50}$ in cell culture (*Kurt Yilmaz et al., 2009*), and therapy failure with resistance mutations for therapies involving DRV are rare and most often occur in people who had previously failed therapy with other protease inhibitors with drug resistance. An examination of the Stanford University HIV Drug Resistance Database (https://hivdb.stanford.edu/) shows a catalog of approximately 25,000 reported HIV-1 sequences with at least one protease resistance-associated mutation, with the majority of these linked to some level of phenotypic resistance to DRV. For sequences associated with a greater than 200-fold change in resistance to DRV, there were an average of 5.4 resistance-associated mutations (with a maximum of seven mutations). Thus, these levels approach the levels of resistance we have obtained and highlight the potential for creating an even higher genetic barrier to resistance in vivo.

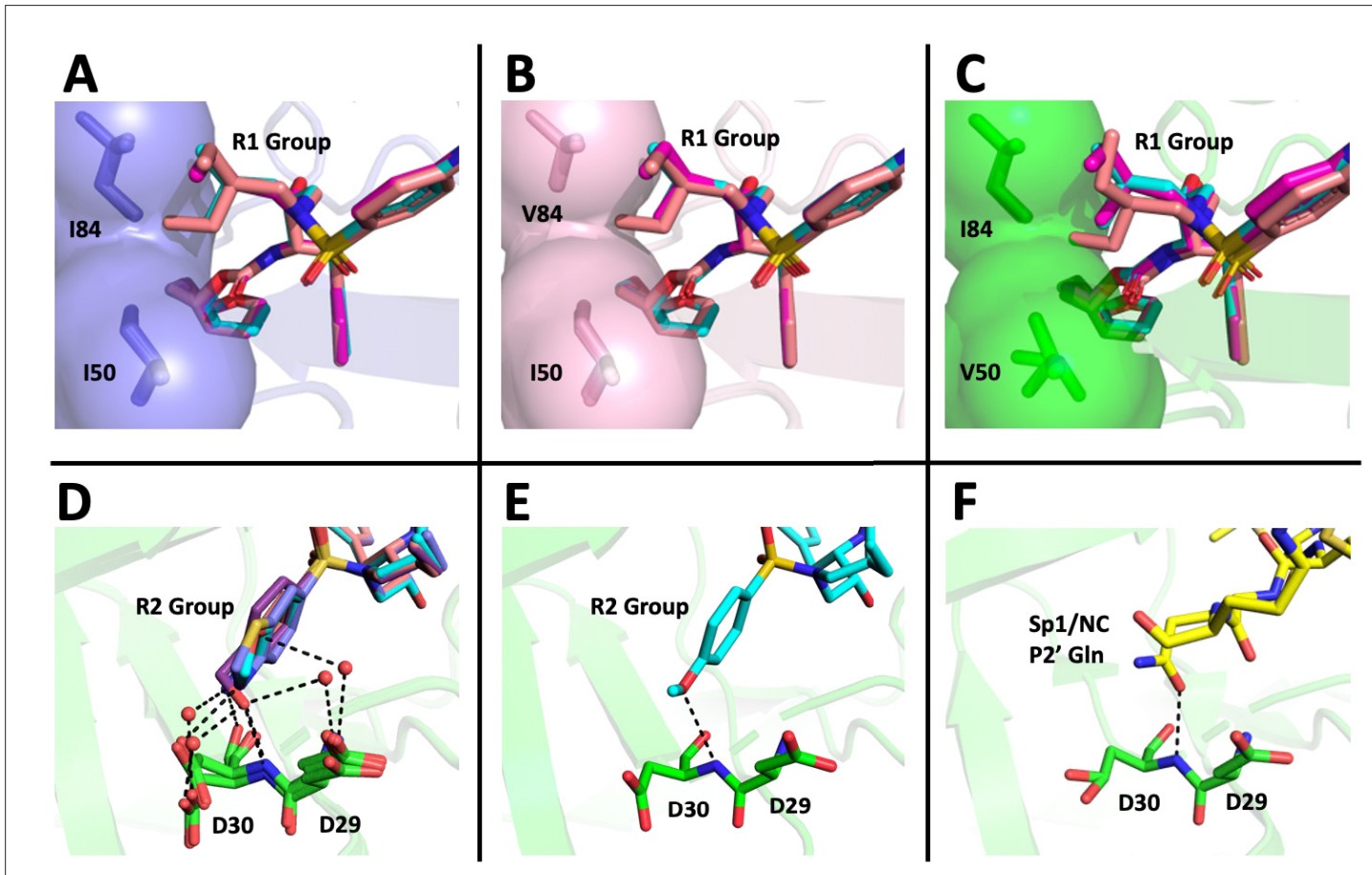

**Figure 8.** Structural interpretation of protease inhibitor resistance and of residual inhibitor potency. (**A–C**) Hydrophobic packing in the S1' subsite in complex with R1 structural groups; darunavir is shown in cyan, UMASS-1 in magenta, and UMASS-6 in salmon. (**A**) The two forms of R1 and darunavir (DRV) are shown packing against wild-type protease at I84 and I50 DRV, UMASS-1, UMASS-6 in wild-type protease (PDB: 6DGX [cyan], 6DGY [magenta], 6DGZ [salmon]). (**B**) Those same inhibitors packing against the I84V mutant (PDB: 6DH0 [cyan], 6DH1 [magenta], 6DH2 [salmon]). (**C**) Those same inhibitors packing against the I50V protease variant (PDB:, 6DH6 [salmon], 6DH7 [magenta], 6DH8 [salmon]). (**D–F**) Binding interactions with Asp29 and Asp30 of the protease S2' subsite in complex with R2 structural groups. Inhibitor/enzyme interactions are shown with black dashed lines representing hydrogen bonds (<3.0 Å). (**D**) UMASS-1, UMASS-3, UMASS-4, and UMASS-5 in WT protease (PDB: 3O99 [salmon], 3O9B [purple], 3O9C [orange], 3O9D [slate blue]). Water molecules are shown as red dots. (**E**) UMASS-2 in WT protease (PDB: 3O9A). (**F**) SP1/NC S2' subsite representation of peptide substrate complexed with an inactive form of wild-type protease (PDB: 1KJ7) showing an interaction between the P2' glutamine and the protein backbone at Asp29 and Asp30.

## P1'/R1 modification

DRV has a relatively small butyl group at this position that most often selected for an I84V mutation. When we extended this moiety by a single methyl group to (S)-2-methylbutyl, we obtained a similar resistance selection profile. However, by extending it with an additional methyl group to (S)-2-methylbutyl or 2-ethyl-n-butyl create an inhibitor that switched the preferred selection to the I50V pathway. In *Figure 8A–C* are shown representations of structures of a subset of the inhibitors with the different P1'/R1 groups with the wild-type protease, the I84V mutant protease, and the I50V mutant protease. As we have previously discussed (*Lockbaum et al., 2019*; *Mittal et al., 2013*), in these structures it is apparent that the P1'/R1-1 group is directed at the protease I84 sidechain, consistent with the shortening of this sidechain as a major resistance mechanism. Conversely, the longer P1'/R1-2 group is oriented into the space between I84 and I50, thus drawing in I50V as the pathway to resistance. As noted in the longitudinal analysis of the selection pathway, the I50V resistance pathway presents additional challenges to the virus in the more rapid accumulation of compensatory mutations (*Figure 4*). Under the circumstances of rapidly declining viral load during the initiation of therapy, this need for additional mutations would represent an enhanced genetic barrier.

## PR2'/R2 modification

The inhibitors with two of the R2 modifications (R2-2 and R2-5) showed greater potency against the mildly resistant proteases (*Figure 2—figure supplement 2*) and against the wild-type virus (*Table 1*). This effect was also evident as a trend for inhibitors with the R2-2 group against the highly resistant proteases (*Figure 5E*) and the highly resistant viruses (*Figure 6E*). When we examined the interactions between protease and inhibitor, there is a clear structural explanation (*Figure 8D–F*). The less potent inhibitors interact with the protease side chains and backbone at D29 and D30 through a network of water molecules. In contrast, with the more potent R2-2 group the water molecules are replaced with direct interactions between inhibitor and protease in the S2' subsite (*Figure 8E*). This direct interaction more closely mimics the interaction made by the glutamine P2' side chain in the optimized protease cleavage site at the SP1/NC boundary (*Figure 8F*). Thus, the ability to replace the water-mediated interactions with a more rigid framework that is interacting with the protease backbone deep in the S2' pocket appears to be a unique feature of the inhibitors with R2-2. Improved potency in another series of PIs was also reported for this R2-2 P2' moiety (*Zhu et al., 2020*).

While the different R2 structures did not select for any novel resistance mutations, we did note that the two inhibitors with the greatest potency as enzymes (UMASS-2 and UMASS-5) were the only inhibitors that selected for I84V and I50V linked on the same genome (*Figure 2A*). This represents a third distinct outcome since in these cases the smaller methylbutyl R1-1 is also able to engage I50. As noted above, the R2-2 group is directly anchored to the protease backbone. We examined the effect this has on the R1-1 orientation. However, we did not see any difference in the placement of the R1-1 group when comparing UMASS-1 (which preferentially selects for I84V) and UMASS-2 (which selects for linked I84V/I50V). It is possible that anchoring to the backbone with R2-2 for UMASS-2 makes the R1-1 group less mobile, reducing the magnitude of the effect of the I84V mutation allowing the inhibitor to maintain some level of interaction with I50, although this inference was not tested directly.

## Protease mutation networks and compensatory mutations in the resistant pathways

In most of the selections, either the I84V or the I50V mutation was largely fixed by the time the inhibitor concentration reached 10 nM in the culture (*Figure 3*, *Figure 3—figure supplement 2*). The viral cultures went through cyclical changes in population diversity as subsequent mutations were added (*Figure 3A*). Because of the large number of selections, it was possible to pool data and see trends in the way mutations were added in these two pathways. Resistance mutations that were detected in multiple selections summed to at least 16 positions (*Figure 2A*). While these mutations are well known, their relationships to each other, and as a function of selective pressure, are less well understood. The data in *Figure 4* show strong linkage between I84V and V32I, while I50V shows linkage with I47V and F53L. These linked compensatory mutations are each close to the primary resistance mutation, suggesting they interact directly with the primary resistance mutation to adjust its position within the subsite to limit interaction with the inhibitor and/or improve interaction with the substrate.

Other mutations appeared often in both pathways but could be found earlier and/or more frequently in one pathway: in the I84V pathway G16E and I54L were favored; in the I50V pathway L10F, I13V, L33F, M46I, L63P, and A71V were favored. In contrast to the linked compensatory mutations, the shared compensatory mutations are often more distant in the structure from the primary resistance mutation. Mutations distal from the active site modulate the enzymatic activity and fitness by altering the dynamic ensemble (*Foulkes-Murzycki et al., 2007*; *Henes et al., 2019*; *Leidner et al., 2021*; *Ragland et al., 2014*; *Ragland et al., 2017*; *Whitfield et al., 2020*) of the enzyme (i.e., conformations sampled). While these changes can sometimes enhance enzymatic activity, they also often significantly contribute to high levels of resistance (*Henes et al., 2019*; *Matthew et al., 2021*) as is observed in these selections. As can be seen in *Figure 2E*, there is a strong correlation between increasing resistance (lower $K_i$) and lost catalytic efficiency (larger kcat/$K_m$). This suggests that compensatory mutations may attenuate fitness loss but do not restore it, or they impact resistance, or both. The role of mutations outside of the active site has been of interest for a long time (*Foulkes-Murzycki et al., 2007*; *Kovalevsky et al., 2006*; *Louis et al., 2011*; *Ode et al., 2006*; *Shen et al., 2010*). In *Figure 4B*, it can be seen that the same compensatory mutation is part of two different resistance pathways but with differing selective pressure to add the mutation, hinting at the complexity of the role of such mutations and their pathway-specific contributions.

In summary, using a number of lengthy selections for resistance to a series of structurally related PIs we have been able to identify the size of the hydrophobic chain at the P1'-equivalent position of the inhibitor as the major determinant for selection of the I84V pathway with a smaller P1'-equivalent chain, including for DRV, versus a larger P1'-eqivalent hydrophobic chain driving selection of the I50V pathway. Also, one of the P2' groups tested gave a higher level of potency even in the face of high-level resistance by creating a direct interaction with the protease backbone. This also led to coselection of I84V and I50V as linked mutations. These chemical changes in the inhibitor increase the genetic barrier for the evolution of resistance and emphasize the potential utility of what a fifth-generation HIV-1 PI could add to regimens with reduced drug complexity.

DRV has been used as the starting point for other approaches in enhancing potency (*Matthew et al., 2021*). These other studies have examined similar, but not identical, structural changes to DRV as described here. There is a cautionary note in comparing between different inhibitors that can be seen in *Figure 2—figure supplement 3*, where the nature of the chemical structure at one site (R2) reverses the order of potency of structures at another site (R1) in the inhibitor. Thus, comparisons between DRV derivatives where there are two or more differences may be problematic in terms of trying to infer parallels or differences. The original description of DRV included a compound with the same R2-2 that was also highly potent (*Surleraux et al., 2005*), and R2-2 has been tested in the context of other changes in DRV (*Delino et al., 2018*). R2-4 has been tested on the DRV backbone (GRL-98065) that was further modified at the P1-equivalent position to give the inhibitor brecanavir (*Amano et al., 2007*; *Hazen et al., 2007*). The R2-5 P2'-equivalent has previously been shown to have interactions with PR amino acids 29/30 in the S2' subsite (*Bulut et al., 2020*). The DRV backbone has also been used as the basis for more extensive chemical changes giving rise to a number of potent HIV-1 protease inhibitors (*Aoki et al., 2017*; *Cihlar et al., 2006*; *Ghosh et al., 2008*; *Ghosh et al., 1998*; *Ghosh et al., 2020*; *Ghosh et al., 2018a*; *Ghosh et al., 2006*; *Ghosh et al., 2018b*; *Ghosh et al., 2017*; *Hazen et al., 2007*; *Miller et al., 2006*; *Miller et al., 2005*; *Miller et al., 2004*; *Nalam et al., 2013*; *Rusere et al., 2019*).

DRV represents a potent antiviral due in large part to its ability to achieve drug levels in the blood that are far above the $EC_{50}$ and the fact that high-level resistance requires many mutations (a high genetic barrier). However, we have shown that resistance usually follows the less deleterious I84V pathway. The UMASS-2 inhibitor anchors the R2-2 group to the protease backbone and allows the intermediate-sized R1-1 group to co-select both the I84V and I50V pathways, creating an even higher genetic bar. These improvements could become important if protease inhibitors are moved to dosing using a long acting depot where drug mass becomes limiting. The enhanced potency of UMASS-2 could potentially be traded for reduced maximum drug levels allowing a given mass of drug to provide antiviral coverage for a longer period of time.

## Materials and methods

### Cell lines and viruses

CEMx174 cells were maintained in RPMI 1640 medium with 10% fetal calf serum and penicillin-streptomycin. TZM-bl and 293T cells were maintained in Dulbecco's modified Eagle-H medium supplemented with 10% fetal calf serum and penicillin-streptomycin. A wild-type virus stock NL4-3 was prepared by transfection of the pNL4-3 plasmid (purified using the QIAGEN Plasmid Maxikit) into HeLa cells. For the mixture of isogenic mutant viruses, the following NL4-3 variants were created, each with a single mutation in the protease with this mixture forming the virus pool for the initiation of selection with mutant viruses: L10I, K20R, K20I, L24I, D30N, V32I, M36I, M46I, M46L, I47V, G48V, F53L, I54V, I62V, L63P, A71T, A71V, G73S, V77I, V82A, V82T, I84V, N88D, N88S, L90M, I93L (*Henderson et al., 2012*). The cell lines used in this study came from the NIH HIV Reagent Program or ATCC, and were initially expanded and frozen down. Cells from these low-passage frozen stocks were thawed and used in experiments, and typically replaced within 1 y. Authentication of cell lines was based on assessments of the providers. Cell phenotypes were monitored for their ability to generate virus after transfection (293Ts), support viral replication (CEMx174), or report viral infection (TZM-bl). The following cell lines were obtained through the NIH HIV Reagent Program, Division of AIDS, NIAID, NIH: 174xCEM Cells, ARP-272, contributed by Dr. Peter Cresswell and TZM-bl Cells, ARP-8129, contributed by Dr. John C. Kappes, Dr. Xiaoyun Wu and Tranzyme Inc. The HEK-293Ts cells were obtained from ATCC, CRL-11268, and used directly upon receipt to created an expanded stock. Certificates of analysis are available on ATCC to include HEK293Ts. Cells obtained from the frozen stock were periodically tested for mycoplasma and were consistently negative.

### Selections

An aliquot of $3 \times 10^6$ CEMx174 cells was incubated at 37°C for 2–3 hr with 250 µl of a virus stock generated from the HIV-1 infectious molecular clone pNL4-3. The culture volume was then brought to 10 ml with RPMI medium. Each flask received one of the following inhibitors at escalating concentrations: UMass1, UMass2, UMass3, UMass4, UMass5, UMass6, UMass7, UMass8, UMass9, UMass10, DRV, and no drug (ND). After 48 hr and every 48 hr after, the cells were pelleted by centrifugation and 10 ml of fresh medium and inhibitors were added. When the culture had undergone extensive cytopathic effect (CPE) indicative of viral replication, the supernatant medium and the cells were harvested separately and stored at –80°C. The virus-containing supernatant was used to start the next round of infection, and after several rounds at the initial concentration, the inhibitor concentration was increased 1.5-fold at each subsequent round of virus passage. The level of resistance (50% inhibitory concentration [$EC_{50}$]) of the single inhibitor-selected virus pools was determined by a TZM infection assay in which the PI is added to productively infected cells and the titers of supernatant virus made in the presence of the inhibitor are determined.

### TZM infection assay

PI dilutions were prepared by taking 10 µM stocked and performing a fivefold serial dilution using RPMI media (final drug concentration is 100 µM). One dilution of drug was added to each well of a 24-well plate and repeated so each virus would have a full set of dilutions. Viruses for the assay were made by seeding $3 \times 10^6$ CEM cells in a 24-well plate and incubating with 250 µl of virus at 37°C for 2–3 hr before bringing the culture to 10 ml with RPMI media. After 48 hr, the medium was changed and repeated every 48 hr after until the culture had undergone CPE. Infected CEM cells were collected and diluted so that 1 ml of cells could be plated in each well containing a unique drug dilution. Then 24 hr later the virus supernatant was collected from each well followed by filtering through a 0.45 µM filter then placed in –80°C. Viruses were thawed and added to 96-well plates in triplicate. TZM-bl cells were collected and diluted to a concentration of $2 \times 10^5$ cells/ml, 100 µl were added on top of the pre-plated viruses. Plates were kept in 37°C, 5% $CO_2$ in an incubator for 48 hr. After 48 hr, the cells in the plates were lysed by removing the medium, washing two times with 100 µl PBS, and then lysed with 1× lysis buffer (made from 5× Promega Firefly Lysis Buffer). Plates were frozen for at least 24 hr and then thawed for 2 hr before analyzing with Promega Firefly Luciferase Kit on a luminometer. Data was analyzed with Prism 7 to fit sigmoidal dose–response curves.

## DNA preparation and amplification of the protease-coding region

Total cellular DNA was isolated from infected cell pellets by using the QIAamp blood kit (QIAGEN). The protease-coding domain of viral DNA was amplified by nested PCR. The PCR conditions are available upon request. PCR products were purified by using QIAquick PCR purification kit (QIAGEN) and directly sequenced or cloned into the pT7Blue vector (Novagen) and sequenced.

## Primer-ID deep sequencing of viral RNA

We used the PID protocol to prepare MiSeq PID libraries with multiplexed primers. Viral RNA was extracted from plasma samples using the QIAamp viral RNA mini kit (QIAGEN, Hilden, Germany). Complementary DNA (cDNA) was synthesized using a cDNA primer mixture targeting protease (PR) with a block of random nucleotides in each cDNA primer serving as the PID, and SuperScript III RT (Thermo Fisher). After two rounds of bead purification of the cDNA, we amplified the cDNA using a mixture of a forward primer that targeted the upstream coding region, followed by a second round of PCR to incorporate the Illumina adaptor sequences. Gel-purified libraries were pooled and sequenced using the MiSeq 300 base paired-end sequencing protocol (Illumina). The sequencing covered the HIV-1 PR region (HXB2 2648-2914, 3001-3257).

We used the Illumina bcl2fastq pipeline for the initial processing and constructed template consensus sequences (TCSs) with TCS pipeline version 1.33 (https://github.com/SwanstromLab/PID) (*Zhou, 2019*). We then aligned TCSs to an HXB2 reference to remove sequences not at the targeted region or that had large deletions. We used the Entropy tool at LANL (https://www.hiv.lanl.gov/content/sequence/ENTROPY/entropy.html) to calculate entropy for each specimen. The sequencing data is available at NIH Sequencing Read Archive (SRA) under BioProject ID PRJNA853351.

## Protease expression and purification

The highly mutated, resistant, protease variant genes were purchased on a pET11a plasmid with codon optimization for protein expression in *Escherichia coli* (Genewiz). A Q7K mutation was included to prevent autoproteolysis (*Rosé et al., 1993*). The expression, isolation, and purification of WT and mutant HIV-1 proteases used for enzymatic assays were carried out as previously described (*Henes et al., 2019*; *King et al., 2002*; *Özen et al., 2014*). Briefly, the gene encoding the desired HIV-1 protease was subcloned into the heat-inducible pXC35 expression vector (ATCC) and transformed into *E. coli* TAP-106 cells. Cells grown in 6 l of Terrific Broth were lysed with a cell disruptor twice, and the protein was purified from inclusion bodies (*Hui et al., 1993*). Inclusion bodies, isolated as a pellet after centrifugation, were dissolved in 50% acetic acid followed by another round of centrifugation at 19,000 rpm for 30 min to remove insoluble impurities. Size-exclusion chromatography was carried out on a 2.1 l Sephadex G-75 Superfine (Sigma Chemical) column equilibrated with 50% acetic acid to separate high molecular weight proteins from the desired protease. Pure fractions of HIV-1 protease were refolded using a tenfold dilution of refolding buffer (0.05 M sodium acetate at pH 5.5, 5% ethylene glycol, 10% glycerol, and 5 µM DTT). Folded protein was concentrated to 0.5–3 mg/ml and stored. The stored protease was used in $K_M$ and $K_i$ assays.

## Enzymatic assays

### $K_m$ assay

$K_m$ values were determined as previously described (*Henes et al., 2019*; *Lockbaum et al., 2019*; *Matayoshi et al., 1990*; *Windsor and Raines, 2015*). Briefly, a 10-amino acid substrate containing the natural MA/CA cleavage site with an EDANS/DABCYL FRET pair was dissolved in 8% DMSO at 40 nM and 6% DMSO at 30 nM. The 30 nM substrate was 4/5 serially diluted from 30 nM to 6 nM. HIV-1 protease was diluted to 120 nM and, and 5 µl were added to the 96-well plate to obtain a final concentration of 10 nM. Fluorescence was observed using a PerkinElmer Envision plate reader with an excitation at 340 nm and emission at 492 nm, and monitored for 200 counts. A FRET inner filter effect correction was applied as previously described (*Liu et al., 1999*). Data corrected for the inner filter effect was analyzed with Prism7.

### $K_i$ assay

Enzyme inhibition constants ($K_i$ values) were determined as previously described (*Henes et al., 2019*; *Lockbaum et al., 2019*; *Matayoshi et al., 1990*; *Windsor and Raines, 2015*). Briefly, in a 96-well

plate, inhibitors were serially diluted down from 2000 to 10,000 nM depending on protease resistance. All samples were incubated with 5 nM protein for 1 hr. A 10-amino acid substrate containing an optimized protease cleavage site (*Windsor and Raines, 2015*), purchased from Bachem, with an EDANS/DABCYL FRET pair was dissolved in 4% DMSO at 120 µM. Using a PerkinElmer Envision plate reader, 5 µl of the 120 µM substrate were added to the 96-well plate to a final concentration of 10 µM. Fluorescence was observed with an excitation at 340 nm and emission at 492 nm and monitored for 200 counts. Data was analyzed with Prism7.

## Acknowledgements

This work was supported by grants from NIGMS (P01-GM109767, R01-GM135919) and NIAID (R01-AI140970) of the NIH. This research received infrastructure support from the UNC CFAR (P30-AI050410), and the UNC Lineberger Comprehensive Cancer Center (P30-CA016086). We also acknowledge receipt of reagents from the NIH HIV Reagent Program: TZM-bl cells (J Kappes and X Wu, contributors) and CEMx174 cells (P Cresswell, contributor). The support of the UNC High Throughput Sequencing Facility is also acknowledged.

## Additional information

### Funding

| Funder | Grant reference number | Author |
|---|---|---|
| National Institute of General Medical Sciences | 1P01GM109767-01A` | Ean Spielvogel<br>Celia A Schiffer<br>Gordon J Lockbaum<br>Shuntai Zhou<br>Mina Henes<br>Akbar Ali<br>Ellen A Nalivaika<br>Klajdi Kosovrasti<br>Nese Kurt Yilmaz<br>Ronald Swanstrom<br>Amy Sondgeroth<br>Sook-Kyung Lee |
| National Institute of General Medical Sciences | R01-GM135919 | Ean Spielvogel<br>Celia A Schiffer<br>Gordon J Lockbaum<br>Shuntai Zhou<br>Mina Henes<br>Akbar Ali<br>Ellen A Nalivaika<br>Klajdi Kosovrasti<br>Nese Kurt Yilmaz<br>Ronald Swanstrom<br>Amy Sondgeroth<br>Sook-Kyung Lee |
| NIAID | R01-AI140970 | Ean Spielvogel<br>Celia A Schiffer<br>Gordon J Lockbaum<br>Shuntai Zhou<br>Mina Henes<br>Akbar Ali<br>Ellen A Nalivaika<br>Klajdi Kosovrasti<br>Nese Kurt Yilmaz<br>Ronald Swanstrom<br>Amy Sondgeroth<br>Sook-Kyung Lee |

| Funder | Grant reference number | Author |
|---|---|---|
| UNC CFAR | P30-AI050410 | Ean Spielvogel<br>Celia A Schiffer<br>Gordon J Lockbaum<br>Shuntai Zhou<br>Mina Henes<br>Akbar Ali<br>Ellen A Nalivaika<br>Klajdi Kosovrasti<br>Nese Kurt Yilmaz<br>Ronald Swanstrom<br>Amy Sondgeroth<br>Sook-Kyung Lee |
| UNC Lineberger Comprehensive Cancer Center | P30-CA016086 | Ean Spielvogel<br>Celia A Schiffer<br>Gordon J Lockbaum<br>Shuntai Zhou<br>Mina Henes<br>Akbar Ali<br>Ellen A Nalivaika<br>Klajdi Kosovrasti<br>Nese Kurt Yilmaz<br>Ronald Swanstrom<br>Amy Sondgeroth<br>Sook-Kyung Lee |

The funders had no role in study design, data collection and interpretation, or the decision to submit the work for publication.

## Author contributions

Ean Spielvogel, Data curation, Formal analysis, Validation, Investigation, Methodology, Writing - original draft; Sook-Kyung Lee, Gordon J Lockbaum, Data curation, Formal analysis, Validation, Investigation, Methodology, Writing – review and editing; Shuntai Zhou, Data curation, Formal analysis, Validation, Investigation; Mina Henes, Data curation, Validation, Investigation, Methodology; Amy Sondgeroth, Data curation, Validation, Investigation, Methodology, Writing – review and editing; Klajdi Kosovrasti, Data curation, Formal analysis, Validation, Investigation, Writing – review and editing; Ellen A Nalivaika, Resources, Data curation, Formal analysis, Investigation, Visualization, Methodology; Akbar Ali, Nese Kurt Yilmaz, Resources, Data curation, Formal analysis, Supervision, Validation, Investigation, Visualization, Methodology, Writing – review and editing; Celia A Schiffer, Conceptualization, Resources, Formal analysis, Supervision, Funding acquisition, Writing - original draft, Project administration; Ronald Swanstrom, Conceptualization, Formal analysis, Supervision, Funding acquisition, Writing - original draft, Project administration, Writing – review and editing

## Author ORCIDs
Celia A Schiffer http://orcid.org/0000-0003-2270-6613
Ronald Swanstrom http://orcid.org/0000-0001-7777-0773

## Decision letter and Author response
Decision letter https://doi.org/10.7554/eLife.80328.sa1
Author response https://doi.org/10.7554/eLife.80328.sa2

# Additional files

## Supplementary files
• Transparent reporting form

## Data availability
The sequencing data (Figure 2, 3, and 4) is available at NIH Sequencing Read Archive (SRA) under BioProject ID PRJNA853351.All source data files for enzymatic Ki and Km (Table 1, Figure 2 and 5) have been uploaded to the Carolina Digital Repository: Swanstrom, Ron, and Ean Spielvogel. Km and Ki Dataset for Selection of Hiv-1 for Resistance to Fifth Generation Protease Inhibitors Reveals Two Independent Pathways to High-level Resistance. 2022.All source data files for EC50 inhibition curves

(Figure 2 and 6) have been uploaded to the Carolina Digital Repository: Swanstrom, Ron, and Ean Spielvogel. Ec50 Dataset for Selection of Hiv-1 for Resistance to Fifth Generation Protease Inhibitors Reveals Two Independent Pathways to High-level Resistance. 2022.

The following datasets were generated:

| Author(s) | Year | Dataset title | Dataset URL | Database and Identifier |
|---|---|---|---|---|
| Henes M, Kosovrasti K, Nalivaika EA, Kurt Yilmaz N, Schiffer CA | 2022 | Km and Ki Dataset for Selection of HIV-1 for Resistance to Fifth Generation Protease Inhibitors Reveals Two Independent Pathways to High-Level Resistance | https://doi.org/10.17615/rb7r-tx62 | Carolina Digital Repository, 10.17615/rb7r-tx62 |
| Spielvogel E, Sondgeroth A, Swanstrom R | 2022 | Display all details of EC50 Dataset for Selection of HIV-1 for Resistance to Fifth Generation Protease Inhibitors Reveals Two Independent Pathways to High-Level Resistance | https://doi.org/10.17615/3697-3v58 | Carolina Digital Repository, 10.17615/3697-3v58 |
| Spielvogel E, Zhou S, Swanstrom R | 2022 | Sequencing Data for: Selection of HIV-1 for Resistance to Fifth Generation Protease Inhibitors Reveals Two Independent Pathways to High-Level Resistance | https://www.ncbi.nlm.nih.gov/bioproject/PRJNA853351 | NCBI BioProject, PRJNA853351 |

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
