## [Editor Report]

This work provides a fundamental understanding of the evolution of resistance to Darunavir, an exceptionally potent HIV-1 protease inhibitor. Conclusions are supported by compelling evidence that small changes in the inhibitor lead to two resistance pathways, each anchored by a specific mutation. These results provide the first evidence for de novo pathway selection and provide an atomic basis for designing the next generation of HIV-1 protease inhibitors.

---

## [Decision Letter]

**Decision letter after peer review:**

Thank you for submitting your article "Selection of HIV-1 for Resistance to Fifth Generation Protease Inhibitors Reveals Two Independent Pathways to High-Level Resistance" for consideration by *eLife*.

Your article has been reviewed by 3 peer reviewers, and the evaluation has been overseen by a Reviewing Editor Owen Pornillos and Sara Sawyer as the Senior Editor. The following individual involved in the review of your submission has agreed to reveal their identity: Dmitry Lyumkis (Reviewer #1).

Essential revisions:

For publication in *eLife*, the paper needs to be revised in order to improve readability (for both experts and non-experts) and to address a number of issues brought up by the Reviewers.

1) Readability. (a) Please include in the introduction a brief "guide" to the P1', P2' positions. (b) Please utilize the *eLife* system of figures and figure supplements (see published examples) to consolidate the main figures and supplementary figures. (c) Presentation of selection data in e.g., Figures1, 3, and S4 need to be improved, for example using a mutational "tree" diagram that illustrates relationships between the primary and accessory mutations.

2) Discussion and data interpretation. (a) The Reviewers are all in agreement that there is more to be said about the accessory variants, how these arise, and how these could contribute to resistance, beyond describing their linkages to the primary mutations. This is key to the paper, but unfortunately, the data are presented in a way that limits comprehension, even by experts. Please determine a more effective way to illustrate (perhaps something similar to an evolutionary tree) and discuss the concepts of key pathways (I84V and I50V), their establishment during selection, and the further addition of and roles of accessory mutations. (b) Address how insights from this study explain the clinical utility and limitation of DRV. (c) Going beyond DRV, discuss design strategies for minimizing drug resistance, including limits on absolute sizes of R1 and R2. (d) Address the potential roles of cleavage site mutations in Gag.

*Reviewer #1 (Recommendations for the authors):*

My comment would be to provide some broader set of rules through which resistance develops, starting with the WT, beyond the 184V vs 150V variants themselves and the pathway switch mediated by modifications at the P1' region, in particular for the accessory mutations. The authors have generated an extensive amount of data from the passaging experiments, using multiple different drugs. Is it possible to devise some conglomerate ordering for resistance onset? Or place the accessory mutations into a broader context? For example, looking at the data in Figure 2, it appears that V82I and M46I somewhat consistently emerge in the initial or mid-stages of the passaging experiments, and they're almost universally present in Figure 1. I47V and M71V, which are also nearly universally present in Figure 1, appear to be more variable in Figure 2, occasionally arising toward the end of the passaging experiment. Are there any other patterns that emerge?

*Reviewer #2 (Recommendations for the authors):*

– The authors report an extended analysis of the development of resistance mutations in HIV protease sequence variation during long-term treatment with darunavir. Long–term drug therapy may also select for mutations in cleavage sites in gag/pol where protease cleaves. Have the authors investigated the presence of cleavage site mutations in the gag sequence using single genome amplification (SGA) or the like? Are all of the changes responsible for resistance attributable to changes in protease only?

– In Figure 1A, the authors use NGS with primer ID to quantify the abundance of drug-resistant variants. These studies are well done and represent the gold standard for measuring levels of different variants. As they show, the highly resistant variants may predominate but other variants are present as well, in some cases seems like these other variants are at substantial levels, and additional discussion of these data would be useful. For instance, in studies of UMASS–2, the most abundant variant is only 37% of the total; this observation seems consistent with the data in figure 2A, where the authors find relatively high Shannon entropy for UMASS-2, reflecting considerable genetic diversity. In contrast, in studies of UMASS-8, where the most abundant variant is also low (only 55.4%), figure 2A shows a relatively low shannon entropy. Were there other closely related variants without DRMS that would explain both the relatively low abundance of resistant variants, but also explain the low entropy levels? Additionally, in these long–term experiments, were there any WT variants present after prolonged drug exposure?

Elsewhere, the authors suggest the presence of widespread recombination in these cultures – might recombination also explain some of the variations in Figure 1A? Considering several log decreases in infectivity demonstrated in Figure 1 D, is recombination contributing to ongoing transmission during drug exposure?

– Figure 2A – Regarding the Shannon entropy determinations – were parallel studies done with darunavir? if so, how do those shannon entropy profiles compare with those measured for UMASS compounds and can they be included in the figure?

– The authors have important new insights in their structural studies – can they place boundaries on the absolute sizes of R1 or R2 for future drug design?

– Previously these authors have provided useful models of genetic variation in vitro (Boucher et al., 2019), including estimating selection coefficients. Can the authors discuss their new data in light of these modeling studies and identify selection coefficients for these variants?

– Do the data from these studies provide insight to explain the clinical utility and limitations of Darunavir as an initial or simplification therapy? For instance:

a) the two–drug combination of Darunavir and Raltegravir is highly effective in suppressing HIV, but this combination is not recommended in individuals with HIV viral loads exceeding 100,000 copies/ml plasma, because of the increased risk for the development of resistance.

b) individuals with baseline protease resistance simplifying their regimens to darunavir+raltegravir did not result in increased frequency of protease resistance (Vizcarra et al., 2019).

– Can you clarify the tinting in Figure S2?

As written:

"UMASS^-1^, -3, -6, -8 show less potency towards these variants and is shaded in a red tint matching R2 groups. Are the tints supposed to be the same for R2 groups?"

But in my version:

UMASS 1 and 3 have an identical tint, but different R2 groups, and

UMASS 6, and 8 have identical tint but different R2 groups.

In table 1

UMASS^-1^ and -6 have identical R2 groups

UMASS-3 and -8 have identical R2 groups

also

UMASS-2 and 5 have identical tint but different R2 groups

UMASS-7 and 10 have identical tint, but different R2 groups

UMSS 10 and 5 have identical R2 groups

UMASS 2 and 7 have identical R2 groups

It's not clear whether this could be an ADOBE issue or something else is meant by this text

and figure.

–Line 474 "demonstrate" should be "demonstrated".

*Reviewer #3 (Recommendations for the authors):*

The manuscript is comprehensive, well–written, and informative, yet dense and with some figures that readers may not find informative. Comments are discussed below.

1. Perhaps it was just me, but I found that figure layout and organization difficult to get the necessary take–home points. In table 1, it might be helpful for the reader to have viral EC50s for the initial mutations, as shown, and the endstage predominates viral populations as well. I am aware that in some cases the final population is a viral mixture of two genotypes and this should be noted.

2. Given that a major focus of the report is mutational selections starting from wildtype from DRV and UMASS compound series to the ending of the selection. The information for the viral mutations arising over time and their respective abundance over time ( = increased drug conc.) is scattered (Figure 1, Sup Figure 4) throughout the Figures. Readers might appreciate a timeline figure documenting when the first mutations arose and their genotypic abundance over time. The primer ID strategy should allow relative quantification of sequences. This is brought up to give the interesting mutational, and somewhat distinct, population pathways in some of the compound selections. Getting at the question of the history of the mutational series is informative. Perhaps a mutational tree?

Figure 1 In the case when both I84V and I50V mutational series are present for (say UMASS–2) do the mutational series occur at the same time? The abundance is for the predominant genotype, what makes up the rest of the genotypic abundance? The data in Figure 2 (and Figure S6) helps to get to this point, but perhaps a single selection figure might be more informative.

3. The authors make the point of the differences in the fitness competency of the I84V and I50V viruses. Were the end viral products of these 2 mutational pathways compared in a head–to–head replication assay in the reported study? This should be addressed.

4. The authors discuss the potential importance of accessory mutations in the I84V and I50V mutational pathway series. Perhaps I missed it, did the authors directly address the contributions of the accessory mutations to resistance, enzymatic function, and improved replication? This should be addressed.

5. The authors point out that given the reported viral fitness loss with the I50V mutation, vs I84V mutation, mutational selection should be skewed towards I50V. Based on the authors' findings UMASS compounds 3, 6–10 appear to favor I50V mutation over I84V, and the I50V accessory series selection. Yet, UMASS 2, 5, 7, and 10 appear the most potent (Table 1 and Figure S2) with resistant viruses. (1). How efficient are the accessory mutations in the I50V mutation series in restoring replication? UMASS compounds 2 and 5 have both I50V and I84V (and accessory mutations) populations, yet the compounds remain potent. Please discuss in the context of next–generation inhibitors.

6. Given the reports of Gag compensatory sites improving viral replication, is there any hint that either of the 2 pathways uses not protease mutational pathways to restore viral fitness? Please discuss.

7. The authors propose using DRV as a template to improve its efficacy. A recent report by Bulut et al. PMID: 32606378 also derives new compounds from DRV. Please discuss.

---

## [Author Response]

Essential revisions:For publication in eLife, the paper needs to be revised in order to improve readability (for both experts and non-experts) and to address a number of issues brought up by the Reviewers.1) Readability.(a) Please include in the introduction a brief "guide" to the P1', P2' positions.

This has been done in the Introduction with a new figure of the structure of darunavir, pointing out the equivalent of each position (P2P2'), the corresponding subsites, and the R1 and R2 ligand positions. Text in the figure legend addresses all of these points of nomenclature at the beginning.

(b) Please utilize the eLife system of figures and figure supplements (see published examples) to consolidate the main figures and supplementary figures.

Done.

(c) Presentation of selection data in e.g., Figures1, 3, and S4 need to be improved, for example using a mutational "tree" diagram that illustrates relationships between the primary and accessory mutations.

For Figure 1 (now Figure 2) we have added the wild type amino acid sequence at the positions relevant to each mutation. This has also been done for Figure S4. Figures 1 (now 2) and S4 are meant to show the endpoint selection and not the intermediate time points so they are otherwise unchanged.

We have added a second panel to Figure 3 (now Figure 4B) that shows the idealized timeline of mutation addition for each pathway. We have also added a supplemental figure to show a "mutation tree" for one selection that used the I84V pathway and one that used the I50V pathway. We have not done this for all of the selections individually as they sometimes fix multiple mutations at one time making the trees not very informative. We would prefer to keep the terminal culture composition and the pathways as two separate discussions as each is complicated. We have reviewed the text and tried to provide small changes to give the reviewer guidance on these points.

2) Discussion and data interpretation.(a) The Reviewers are all in agreement that there is more to be said about the accessory variants, how these arise, and how these could contribute to resistance, beyond describing their linkages to the primary mutations. This is key to the paper, but unfortunately, the data are presented in a way that limits comprehension, even by experts. Please determine a more effective way to illustrate (perhaps something similar to an evolutionary tree) and discuss the concepts of key pathways (I84V and I50V), their establishment during selection, and the further addition of and roles of accessory mutations.

As noted above, we have provided a more accessible visual of when the accessory mutations are added in the new Figure 4B. This figure also separates the mutations that are linked to either I50V or I84V as opposed to those that are common. We now note that the linked mutations are close spatially to the primary mutation while the shared mutations are on average more distal.

We have also added text on this topic based on Dr. Schiffer's published work with several of the sequences identified in these selections (two paragraphs starting at line 689). However, a comprehensive discussion of accessory mutations is a job for a review article. These have been studied using modeling, molecular dynamics, biophysical, biochemical, genetic, and virologic approaches in a myriad of sequence combinations. We agree that understanding accessory mutations is of intrinsic interest in terms of protein function and important in terms of understanding mechanisms of resistance. Our work has clarified the relationship of some of the accessory mutations with the two specific pathways but to go beyond that in a way that would represent a rigorous discussion is beyond the scope of this publication. We appreciate the interest in this subject but we also hope everyone understands the complexity where each accessory protein may work by a different mechanism with common and specific accessory mutations and with larger or smaller effects even when shared between pathways.

(b) Address how insights from this study explain the clinical utility and limitation of DRV.

This point is now covered in the last paragraph of the text where we discuss issues of drug potency and the potential to trade enhanced potency for lower drug levels that could allow dosing in a depot format to allow a protease inhibitor to be used in a long acting formula.

(c) Going beyond DRV, discuss design strategies for minimizing drug resistance, including limits on absolute sizes of R1 and R2.

Our additional analysis, prompted by the instruction not to use "data not shown", has led to an insight we had not appreciated that addresses this question (issues that Reviewer 3 thoughtfully raised). We did not detect new/unique mutations based on the R2 group. However, we (finally) noticed that two of the R2 groups (2 and 5) both reversed the potency associated with size in the R1 group (R1-1 was more potent than R1-2) and with the R1-1 group co-selected for both primary resistance mutations (I50V and I84V). We initially thought this co-selection was a stochastic feature of the data but now see that it is linked both to the most potent inhibitors and to the reversal of R1 potency providing strong evidence of its significance even if the number of observations is low (N=3). Thus it is not the size that appears to be significant but the anchoring of R2 to the protease backbone which enhances the potency of the intermediate-sized R1-1. These points have now been added as a major conclusion in the paper and we are a bit embarrassed we did not see this connection before the first submission but are grateful to have been prompted to do so before publication. We now see that the intermediate size of R1 (bigger than DRV and smaller than R1-2) is optimal when anchoring R2 to the protease backbone. This seems like a good place to leave the interpretation of our data without further speculation.

(d) Address the potential roles of cleavage site mutations in Gag.

We have added text at the end of the Results covering this topic. There is also a new figure covering both the data and the structural interpretation (Figure 7). This analysis had been done but had been cut from the submitted manuscript to save length.

Reviewer #1 (Recommendations for the authors):My comment would be to provide some broader set of rules through which resistance develops, starting with the WT, beyond the 184V vs 150V variants themselves and the pathway switch mediated by modifications at the P1' region, in particular for the accessory mutations. The authors have generated an extensive amount of data from the passaging experiments, using multiple different drugs. Is it possible to devise some conglomerate ordering for resistance onset? Or place the accessory mutations into a broader context? For example, looking at the data in Figure 2, it appears that V82I and M46I somewhat consistently emerge in the initial or mid-stages of the passaging experiments, and they're almost universally present in Figure 1. I47V and M71V, which are also nearly universally present in Figure 1, appear to be more variable in Figure 2, occasionally arising toward the end of the passaging experiment. Are there any other patterns that emerge?

The reviewer has raised an important point and one we want very much to convey. Since we apparently did not do a good job of that we are thankful to the reviewer for raising this issue. We had designed Figure 3 (now Figure 4A) to show this point since it shows both the timing of appearance and the frequency with which each mutation appears when normalized by drug concentration across the different selections; clearly this was not an effective way to convey this information. We have retained this figure since it represents the interpretation of the primary data but we now reproduce this information in two ways. First, there is now a Figure 4B where we present this same information as a timeline with the mutations shown as sequentially entering with arrows, with the size of the arrow representing the frequency with which this happens among the different selections. Also, the figure is drawn to easily distinguish the common versus the unique mutations in the two pathways. Second, in the new supplemental figure linked to Figure 4 we show a phylogenetic tree following the accumulations of mutations in a culture that followed each pathway. While we don't feel this is the most accessible way to present the data we have included it since this is an analytical approach that is familiar to most readers.

Reviewer #2 (Recommendations for the authors):– The authors report an extended analysis of the development of resistance mutations in HIV protease sequence variation during long-term treatment with darunavir. Long–term drug therapy may also select for mutations in cleavage sites in gag/pol where protease cleaves. Have the authors investigated the presence of cleavage site mutations in the gag sequence using single genome amplification (SGA) or the like? Are all of the changes responsible for resistance attributable to changes in protease only?

We had minimized our data on this point in the initial submission given the length requirements. We now present our data on cleavage sites at the end of the Results (Figure 7). Because we did so many selections we have robust genetic data on this point and are able to link protease mutations to cleavage site mutations with models of structures to interpret the results. In addition, we recently "decoded" the sequence preferences of protease cleavage sites and can also interpret both the wild type and mutant sequences in that context.

– In Figure 1A, the authors use NGS with primer ID to quantify the abundance of drug-resistant variants. These studies are well done and represent the gold standard for measuring levels of different variants. As they show, the highly resistant variants may predominate but other variants are present as well, in some cases seems like these other variants are at substantial levels, and additional discussion of these data would be useful. For instance, in studies of UMASS–2, the most abundant variant is only 37% of the total; this observation seems consistent with the data in figure 2A, where the authors find relatively high Shannon entropy for UMASS-2, reflecting considerable genetic diversity. In contrast, in studies of UMASS-8, where the most abundant variant is also low (only 55.4%), figure 2A shows a relatively low shannon entropy. Were there other closely related variants without DRMS that would explain both the relatively low abundance of resistant variants, but also explain the low entropy levels? Additionally, in these long–term experiments, were there any WT variants present after prolonged drug exposure?

We appreciate that the reviewer delved into the large amount of data associated with these selections to bring out this point. First, we did not detect any wild type sequence at the end of the selections. Second, there is no simple "this is it" answer to questions about all specific details of the selections. Each selection is different with what are probably some stochastic bottlenecks that are carried forward. In terms of the differences in Shannon Entropy that likely resides in the fact that the top two variants in the UMASS-2 selection (starting with wild type) add to only 71% of the population while for the UMASS-8 selection (starting with wild type) the top two variants account for 97% of the population. It is possible that some of the variability at the end could have been further reduced by several more passages at the highest inhibitor concentration. We assume that the presence of several variants at significant levels represent variants with similar fitness and resistance levels such that longer passage in the presence of drug is needed to allow one of the otherwise similar variants to outgrow the other. The minor variants always have the core mutations that define the pathway and differ by minor mutations that are probably present for a variety of reasons. While some of these minor mutations can sometimes also be present as fixed mutations (such as I13V), other mutations may be deleterious such as the R87K mutation within the highly conserved GRN α helix motif; this mutation likely occurs through APOBEC3G mutagenesis and is carried in the population at a very low level but contributes to low level variability. It is difficult to discuss the data at this level within the text so we are happy that an especially interested reader will be able to read about these points in the published review/response format.

Elsewhere, the authors suggest the presence of widespread recombination in these cultures – might recombination also explain some of the variations in Figure 1A? Considering several log decreases in infectivity demonstrated in Figure 1 D, is recombination contributing to ongoing transmission during drug exposure?

We are unclear as to what the reviewer is asking with this question. We assume recombination is always available to the virus under our culture conditions where all cells are ultimately killed, giving the opportunity for dual infection and packaging of heterodimeric RNA (the precursor to recombination during the subsequent infection). Although we have not measured recombination, we assume it is one of the mechanisms at work in generating genetic diversity on which the drug selective pressure is working.

– Figure 2A – Regarding the Shannon entropy determinations – were parallel studies done with darunavir? if so, how do those shannon entropy profiles compare with those measured for UMASS compounds and can they be included in the figure?

The same deep sequencing was done for the DRV selections so the same information on Shannon entropy is available. This information is now included a supplemental figure.

– The authors have important new insights in their structural studies – can they place boundaries on the absolute sizes of R1 or R2 for future drug design?

We appreciate this comment as it caused us to look again at our data with an eye toward "what's next." However, in doing this we noted that two of the R2 groups reversed the order of potency of the R1 groups and changed the selection pathway to co-select both primary resistance mutations. Thus the anchoring of the R2 group to the protease backbone appears to create an optimal intermediate-sized R1 group. Thus size is not the issue but specific interactions. Since R1-1 and R2-2 appear to interact in an unexpected way it is hard to speculate on how to further improve this outcome.

– Previously these authors have provided useful models of genetic variation in vitro (Boucher et al., 2019), including estimating selection coefficients. Can the authors discuss their new data in light of these modeling studies and identify selection coefficients for these variants?

We appreciate the reviewer challenging us to think more broadly about this question. However, in trying to think through this specific example it seems problematic. In Boucher et al., the metric of fitness was always relative to wild type. In our studies we are continuously changing target fitness sequence with increasing drug concentration. Furthermore, it is likely that we could passage virus forever at 100 nM drug concentration and the virus would settle in a pattern of sequence changes that would be different (i.e. fewer mutations) than present in virus taken to 1 uM. We capture the root of this idea in a qualitative way in the new Figure 4B but the approach that we used in virus passage and the variable way we raised drug concentrations in the many different cultures precludes a meaningful mathematical analysis.

– Do the data from these studies provide insight to explain the clinical utility and limitations of Darunavir as an initial or simplification therapy? For instance:a) the two–drug combination of Darunavir and Raltegravir is highly effective in suppressing HIV, but this combination is not recommended in individuals with HIV viral loads exceeding 100,000 copies/ml plasma, because of the increased risk for the development of resistance.b) individuals with baseline protease resistance simplifying their regimens to darunavir+raltegravir did not result in increased frequency of protease resistance (Vizcarra et al., 2019).

Our work does not speak directly to these points. There has long been an unexplained relationship between reduced drug efficacy and high viral load. It seems likely that this has to do with the population size of the virus in a person, with the larger populations having a higher probability of generating resistance. Therapy failure without resistance is a complex issue. Often this is the result of poor adherence. However, resistance to one of the drugs in a regimen may allow breakthrough replication in the face of the other drug without resistance (at least initially). Our poor understanding of these issues is an obstacle in interpreting data from drug simplification clinical studies.

– Can you clarify the tinting in Figure S2?As written:"UMASS^-1^, -3, -6, -8 show less potency towards these variants and is shaded in a red tint matching R2 groups. Are the tints supposed to be the same for R2 groups?"But in my version:UMASS 1 and 3 have an identical tint, but different R2 groups, andUMASS 6, and 8 have identical tint but different R2 groups.In table 1UMASS^-1^ and -6 have identical R2 groupsUMASS-3 and -8 have identical R2 groupsalsoUMASS-2 and 5 have identical tint but different R2 groupsUMASS-7 and 10 have identical tint, but different R2 groupsUMSS 10 and 5 have identical R2 groupsUMASS 2 and 7 have identical R2 groupsIt's not clear whether this could be an ADOBE issue or something else is meant by this textand figure.

Sorry, we were not very coherent in explaining this figure. The red tinted data points/inhibitors behave differently from the green tinted ones. They come in pairs based on the R1 group (-1 and -6) etc., but two pairs are tinted red and two pairs green because the pairs act similarly by color. We have tried to clarify this language.

–Line 474 "demonstrate" should be "demonstrated".

Corrected.

Reviewer #3 (Recommendations for the authors):The manuscript is comprehensive, well–written, and informative, yet dense and with some figures that readers may not find informative. Comments are discussed below.1. Perhaps it was just me, but I found that figure layout and organization difficult to get the necessary take–home points. In table 1, it might be helpful for the reader to have viral EC50s for the initial mutations, as shown, and the endstage predominates viral populations as well. I am aware that in some cases the final population is a viral mixture of two genotypes and this should be noted.

We have added the endpoint EC50s to Table 1 for those virus pools where it was tested.

2. Given that a major focus of the report is mutational selections starting from wildtype from DRV and UMASS compound series to the ending of the selection. The information for the viral mutations arising over time and their respective abundance over time ( = increased drug conc.) is scattered (Figure 1, Sup Figure 4) throughout the Figures. Readers might appreciate a timeline figure documenting when the first mutations arose and their genotypic abundance over time. The primer ID strategy should allow relative quantification of sequences. This is brought up to give the interesting mutational, and somewhat distinct, population pathways in some of the compound selections. Getting at the question of the history of the mutational series is informative. Perhaps a mutational tree?

As noted above we have now presented this information in two ways:

As an idealized selection timeline in Figure 3b (which we feel is more accessible) and as a "mutational tree" in a supplemental figure for a typical selection.

Figure 1 In the case when both I84V and I50V mutational series are present for (say UMASS–2) do the mutational series occur at the same time? The abundance is for the predominant genotype, what makes up the rest of the genotypic abundance? The data in Figure 2 (and Figure S6) helps to get to this point, but perhaps a single selection figure might be more informative.

The reviewer gets at an interesting point although the sample size for making conclusions is getting too small to speak with a lot of confidence. In the entropy figure (now Figure 3) it appears there are two answers to this question. When we started the culture with many mutants then I84V/V32I come in early and I50V comes in late (UMASS-2). However, when we start with wild type then I84V and I50V come in first together with I47V coming in before V32I, suggesting the I50V is driving the selection for the linked mutation (UMASS-2 and -5). While this is an interesting nuance in the data (and it is present for someone patient enough to go look for it), we feel it is based on too few selections to be able to make a strong conclusion.

3. The authors make the point of the differences in the fitness competency of the I84V and I50V viruses. Were the end viral products of these 2 mutational pathways compared in a head–to–head replication assay in the reported study? This should be addressed.

We have not done a head-to-head comparison. As noted, we have virus pools, not clones. While these pools are sufficient for getting an EC50 in short term replication, assessing fitness requires multiple rounds of replication and should be done with a clone. Realistically, this is another study to tear apart these pathways for the effect of individual mutations and differences in the pathways.

4. The authors discuss the potential importance of accessory mutations in the I84V and I50V mutational pathway series. Perhaps I missed it, did the authors directly address the contributions of the accessory mutations to resistance, enzymatic function, and improved replication? This should be addressed.

We have noted that to the extent there are mutations that are just present for "fitness" they do not restore fitness but rather attenuate the loss of fitness due to resistance (as seen in now Figure 2E). There is a large body of literature on accessory/compensatory mutations, not all of which is consistent and that has been generated using a number of not easily compared experimental approaches. As shown in Figure 4B we can now start to order their appearance and linkage. Dr. Schiffer has started to look at these mutations in biochemical, structural, and molecular dynamics studies and we now more clearly reference that work. However, it is beyond the scope of this study to add further clarity into the roles of these extra mutations.

5. The authors point out that given the reported viral fitness loss with the I50V mutation, vs I84V mutation, mutational selection should be skewed towards I50V. Based on the authors' findings UMASS compounds 3, 6–10 appear to favor I50V mutation over I84V, and the I50V accessory series selection. Yet, UMASS 2, 5, 7, and 10 appear the most potent (Table 1 and Figure S2) with resistant viruses. (1). How efficient are the accessory mutations in the I50V mutation series in restoring replication? UMASS compounds 2 and 5 have both I50V and I84V (and accessory mutations) populations, yet the compounds remain potent. Please discuss in the context of next–generation inhibitors.

The reviewer correctly identifies a feature of our data we had not adequately considered. We have added extensive discussion of this point as we believe UMASS-2 and -5 are more potent due to their R2 group and even reversing the size advantage of R1-2 over R1-1. This enhanced interpretation of our data is now an important part of the paper.

6. Given the reports of Gag compensatory sites improving viral replication, is there any hint that either of the 2 pathways uses not protease mutational pathways to restore viral fitness? Please discuss.

Yes, we had that data but left it out for length considerations. It is now included at the end of the Results section.

7. The authors propose using DRV as a template to improve its efficacy. A recent report by Bulut et al. PMID: 32606378 also derives new compounds from DRV. Please discuss.

We have included a discussion of this (and related) work near the end of the Discussion. It was an oversight on our part not to have included these relevant references.